**TOOLS**

# Systematic analysis of YFP traps reveals common mRNA/protein discordance in neural tissues

Joshua S. Titlow[1]*, Maria Kiourlappou[1]*, Ana Palanca[1]*, Jeffrey Y. Lee[1]*, Dalia S. Gala[1], Darragh Ennis[1], Joyce J.S. Yu[1], Florence L. Young[1], David Miguel Susano Pinto[1], Sam Garforth[1], Helena S. Francis[1], Finn Strivens[1], Hugh Mulvey[1], Alex Dallman-Porter[1], Staci Thornton[1], Diana Arman[1], Marissa J. Millard[1], Aino I. Järvelin[1], Mary Kay Thompson[1], Martin Sargent[2], Ilias Kounatidis[1], Richard M. Parton[1], Stephen Taylor[2], and Ilan Davis[1]

**While post-transcriptional control is thought to be required at the periphery of neurons and glia, its extent is unclear. Here, we investigate systematically the spatial distribution and expression of mRNA at single molecule sensitivity and their corresponding proteins of 200 YFP trap lines across the intact *Drosophila* nervous system. 97.5% of the genes studied showed discordance between the distribution of mRNA and the proteins they encode in at least one region of the nervous system. These data suggest that post-transcriptional regulation is very common, helping to explain the complexity of the nervous system. We also discovered that 68.5% of these genes have transcripts present at the periphery of neurons, with 9.5% at the glial periphery. Peripheral transcripts include many potential new regulators of neurons, glia, and their interactions. Our approach is applicable to most genes and tissues and includes powerful novel data annotation and visualization tools for post-transcriptional regulation.**

## Introduction

Neurons are the most extremely polarized cell type in multicellular organisms with many distinct peripheral sites that have to act independently, namely dendritic and axonal synapses. It has been generally accepted that delivering molecules to the periphery of cells involves mRNA localization in fibroblast cells (Sundell and Singer, 1990). But in neurons, the periphery is a large distance from the cell body, requiring long-distance transport and localized translation of peripheral transcripts to regulate protein levels at the synapses (Holt et al., 2019). Conclusive examples of mRNA transport and localized translation in dendrites and oligodendrocytes have been known for decades (Carson et al., 1998; Steward et al., 1998). However, axonal localization and local translation have been easier to discover in developing axons and slower to be elucidated in mature axons. Nevertheless, convincing examples have been known for some time (Jung et al., 2012), despite mRNA being only found at low concentrations at or near the distant synapses of axons. Efforts to address the relative proportion of mRNAs that are locally translated at synapses have led to some outstanding studies, showing that localized mRNA and local translation are common (Hafner et al., 2019). Such data have been complemented with specific conclusive experiments in intact nervous systems (Wang et al., 2009). However, it is not known what the relative contribution of local translation versus nuclear transcription and protein transport is in the diverse cell types of an intact functional mature nervous system.

A hallmark of post-transcriptional regulation is that the distribution of individual species of protein and mRNA is discordant or uncorrelated. Such discordance is most obviously manifested in a lack of correlation between the levels of mRNA expression and protein levels across distinct cell types in tissue, through mRNA stability differences or variations in the rates of translation. However, post-transcriptional regulation can also manifest itself within a cell so that a protein is localized to a distinct site from the mRNA that encodes it. Many mechanisms can lead to intracellular protein and mRNA discordance, including localized translation, mRNA degradation, or intracellular transport of protein or mRNA (Mofatteh and Bullock, 2017). To date, systematic characterization of discordance between protein and mRNA has not been carried out across a whole intact nervous system or any other complex tissue. The advent of single-cell transcriptomics (Aldridge and Teichmann, 2020) and

[1]Department of Biochemistry, University of Oxford, Oxford, UK; [2]Weatherall Institute for Molecular Medicine, University of Oxford, Oxford, UK.

*J.S. Titlow, M. Kiourlappou, A. Palanca, and J.Y. Lee contributed equally to this paper. Correspondence to Ilan Davis: ilan.davis@bioch.ox.ac.uk

J.S. Titlow's current affiliation is Wellcome Leap - Delta Tissue Program, Los Angeles, CA, USA. A. Palanca's current affiliation is Departamento de Anatomía y Biología Celular, Universidad de Cantabria, Cantabria, Spain. I. Kounatidis's current affiliation is School of Life Health and Chemical Sciences, The Open University, Walton Hall, Milton Keynes, UK. M.J. Millard's current affiliation is Department of Internal Medicine, Wake Forest School of Medicine, Winston Salem, NC, USA.

spatial transcriptomics (Marx, 2021) has been a major transformational step. However, although single-cell proteomics and high-coverage imaging mass spectrometry are on the horizon (Marx, 2019), the methods currently lack sufficient sensitivity or coverage, have limited resolution, and are unable to multiplex RNA and protein detection at substantial scale within the same cell. Given the extremely low copy number of mRNA in the periphery of axons and dendrites and their small diameter, current spatial transcriptomics technologies, such as Nanostring, lack both resolution and sensitivity for systematic spatial characterization of transcriptomes in intact nervous systems. Moreover, single-cell transcriptomics approaches lose the peripheral compartments of cells, so are not applicable to systematically address peripheral localization in neurons. Single-molecule FISH methods such as merFISH can overcome the issues of resolution and sensitivity but are not compatible with systematic spatial protein analysis.

Here, we have overcome these technical limitations by developing a widely applicable workflow for comparing the level of discordance between hundreds of mRNAs and their corresponding proteins at high resolution across complex tissues in 3D. Our approach depends on the use of a fluorescent protein to tag many individual endogenous genes and systematic visualization of mRNA using single-molecule FISH (smFISH) to detect >78% of all individual tagged molecules of mRNA in every cell at high resolution. mRNA detection is coupled with covisualization of protein at high sensitivity and resolution in the same specimens. We have prototyped our workflow for 200 genes in the intact nervous system and neuromuscular junction (NMJ) of third-instar *Drosophila* larva using a collection of YFP fusions (Lowe et al., 2014). Our unexpected results led us to a wholesale revision of the global view of post-transcriptional regulation, mRNA localization, and delivery of proteins to the periphery of the nervous system. Post-transcriptional regulation is very common across all of the nervous systems, acting hand in hand with transcriptional regulation to create a complex tapestry of protein distribution in time and space. We present our data as a resource that is easily browsable in the context of a rich landscape of genomics, functional, and bioinformatics data using Multi-Dimensional Data Viewer (MDV), an open-source and flexible software platform (Weeratunga et al., 2022 *Preprint*).

## Results

### Systematic analysis of the level and distribution of the mRNAs of 200 genes by smFISH against YFP fusions and their corresponding fluorescent protein visualization across the nervous system

To ask how gene expression is controlled in specific cell types and subcellular compartments in the nervous system, we developed an imaging pipeline to simultaneously quantify transcription, mRNA, and protein levels throughout whole tissues for hundreds of different genes (Fig. 1 A). Our scalable approach takes advantage of *Drosophila* gene trap collections that have a fluorescent protein reporter inserted into introns of individual genes, flanked with splice donor and acceptor sites. Using a common smFISH probe against the mRNA sequence encoding YFP, we detected reporter mRNAs along with an encoded

reporter YFP protein. The smFISH probe also acts as a transcription reporter by detecting primary transcripts at the endogenous gene locus in nuclei. Imaging the smFISH probe and fluorescent protein tag in whole tissues with confocal microscopy allowed us to systematically map the spatial distribution of gene expression in many different regions and cells of the nervous system at high sensitivity and resolution.

As proof of principle, we performed smFISH experiments on a Discs large 1 protein trap line (Dlg1::YFP) in the larval central brain (Fig. 1, B and C). *Dlg1* (PSD95 in mammals) is a tumor suppressor gene encoding a protein that localizes to intercellular junctions (Peng et al., 2000; Albertson and Doe, 2003). Our YFP probe set produced punctate signals typical of individual transcripts that were diffraction-limited spots of uniform intensity and 3D fluorescence intensity distributions (Fig. 1, D and E; Raj et al., 2008; Yang et al., 2017; Titlow et al., 2018). These consistent characteristics of the punctae allow us to easily distinguish true single molecules from discrete background fluorescence shapes that are either larger than diffraction-limited spots or have lower intensity than the single molecules (Fig. 1, E–N, and Fig. S1 A). We found that our YFP smFISH probe is highly sensitive and specific for *dlg1::YFP* mRNA and that the reporter insertion does not affect the localization or expression level of the *dlg1* mRNA or protein (Fig. S1, B–E). To assess whether homozygous lethal YFP insertion leads to an overall reduced level of gene expression, we compared the number of nervous system compartments with YFP-fusion protein or mRNA expression between homozygous viable and lethal lines from our scoring. The analysis revealed homozygous viable and lethal lines show comparable numbers of nervous system compartments that express either protein or mRNA (Fig. S1, F–I). Therefore, our data suggest the homozygous lethality is unlikely to skew our expression scoring survey.

To determine the specificity of the YFP smFISH probe, we tested whether the probe detects any transcripts in a wild-type line that lacks YFP (Fig. 1). While in the *dlg1::YFP* gene trap line, the YFP smFISH probe labels hundreds of diffraction-limited punctae throughout the central brain (Fig. 1, D–I), no equivalent signal could be detected in wild-type samples (Fig. 1, J–N). The majority of individual punctae appearing in the *dlg1::YFP* line (85% in the brain, 78% in larval muscles [Fig. 1 E″]) were also detected by a spectrally separated second oligonucleotide probe set targeting the endogenous *dlg1* exon, indicating that the probe is highly sensitive (Fig. 1 D). Importantly, the Dlg1::YFP protein showed its characteristic enrichment at the cell surface, which means that the reporter protein does not disrupt localization or expression level of the endogenous protein. We conclude that this is an effective approach to screen for gene expression patterns and proceeded to apply the method to 200 gene insertions randomly selected from the Cambridge Protein Trap Insertion (CPTI) collection (Lowe et al., 2014). We not only imaged the central brain and neuroblasts (neural stem cells), but also the mushroom body (equivalent to mammalian hippocampus), optic lobe, ventral nerve cord (VNC), segmental nerves, and the larval NMJ neurons, muscles, and associated glia.

To determine how well the selected set of genes captures the diversity of gene expression patterns in the whole transcriptome,

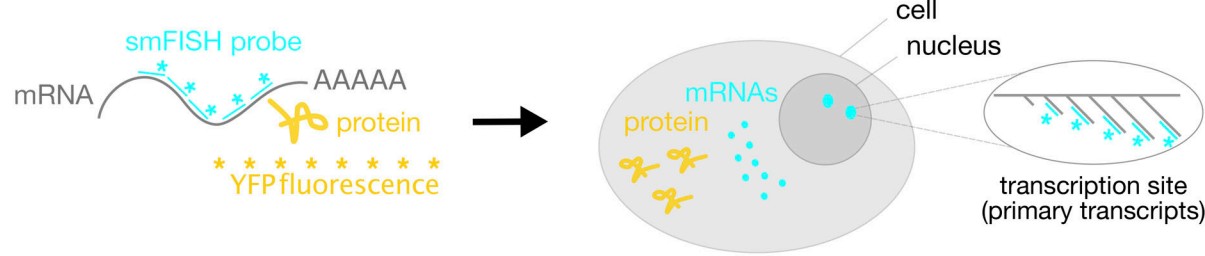

## A  Visualizing RNA and protein localization simultaneously

## B

CPTI002569 YFP insertion site

*dlg1* transcript

YFP smFISH probe     Dlg1 smFISH probe

intron

exon     UTR

4kb

## C

larval CNS     neuroblast

protein localized to the membrane

mRNA in the cytoplasm

nuclei  YFP protein  *dlg1* mRNA  *YFP* mRNA

**D** *dlg1::YFP*  10 µm

**E'**  1 µm

**E''**

Tissue: CNS, NMJ

Co-detection %: 40, 60, 80, 100

YFP mRNA spots

Dlg1 mRNA spots

**E** *dlg1::YFP*  merge  2.5 µm

**F** *dlg1* mRNA

**G** *YFP* mRNA

**H** YFP protein

**I** nuclei

**J** WT CS (-control)  merge  2.5 µm

**K** *dlg1* mRNA

**L** *YFP* mRNA

**M** YFP protein

**N** nuclei

Figure 1. **Spatial detection of localized mRNA and protein expression across multiple tissues and hundreds of different genes in *Drosophila* larvae.** **(A)** General strategy for simultaneously visualizing RNA and protein in fluorescent protein trap lines. RNA is detected using smFISH probes targeting the genetically encoded YFP sequence that is present within the mRNA expressed from the trapped gene. Large transcription foci are seen at the gene locus where there are multiple primary transcripts. Protein is detected by fluorescence of YFP in the protein. This approach can be used to detect any trapped gene in any tissue. Here, we have focused on the nervous system. **(B)** Genetic architecture of CPTI002569, a YFP gene trap line that was used for one set of control experiments. This line has a YFP reporter inserted into an intron that is contained in the *dlg1* transcript. smFISH probes were designed to target the YFP

sequence or a common *dlg1* exon. The probes were labeled with spectrally separated dyes to perform a codetection experiment. **(C)** Schematic showing a region of the *Drosophila* larval CNS that was imaged to visualize *dlg1* expression in neuroblast lineages. The inset schematic shows the endogenous expression pattern of *dlg1* mRNA and protein in a single neuroblast. **(D)** Overview maximum intensity projection confocal image from a *dlg1::YFP* line showing expression of YFP protein (yellow), YFP smFISH probe (cyan), and *dlg1* smFISH probe (magenta). Signal from both smFISH probes was observed as individual diffraction-limited punctae throughout the larval central brain. **(E)** Positive control experiment—high magnification image of the inset in D showing individual transcripts and protein expression within a single cell. **(E′)** Inset, colocalization of YFP and dlg1 smFISH spots, which appear as white pixels to visualize transcripts that were detected by smFISH probes against two different sequences within the dlg::YFP transcript. **(E′′)** Percentage of *yfp* and *dlg1* codetection in both CNS and NMJ tissues. **(F–I)** Grayscale images of the individual channels shown in E. **(J)** Negative control experiment—maximum intensity projection image of the same region in a wild-type line that does express YFP. Note that *dlg1* transcripts are detected by the *dlg1* smFISH probes, but there is no signal in the YFP smFISH channel. **(K–N)** Grayscale images of the individual channels shown in J.

we analyzed published data on the gene expression levels, gene structure, and gene functions. We found that this set of 200 genes provides a fairly representative sample of transcript heterogeneity. We first analyzed publicly available bulk RNA sequencing data from specific tissues and developmental stages. Given that CPTI lines were selected for the presence of YFP reporter expression in embryos, we compared the overall distribution of gene expression levels in embryos to third-instar larval brains. Violin plots show that the distributions of gene expression levels are similar in the two tissues, and by overlaying the individual genes that were screened in the current study (Fig. S2 A), it is clear that the genes we analyzed are relatively abundant and span the entire range of gene expression levels, making the collection a useful proxy for the whole transcriptome.

Next, we characterized the physical structure of the 200 screened genes. The CPTI collection was created by a hybrid *piggyBac* vector insertion which favors longer genes and longer introns since the gene traps are formed by random insertions into introns (Lowe et al., 2014). We found that the 200 CPTI genes we analyzed are indeed as expected on average slightly larger and contain longer introns than the average protein-coding gene (Fig. S2 B). Since it is thought that genes that are highly expressed in the nervous system tend to be longer and contain longer introns than average (McCoy and Fire, 2020), we conclude that our 200 genes are likely to be enriched in genes that are highly expressed in the nervous system.

In contrast, we found that the 3′UTR extension lengths were similar in the 200 CPTI lines compared with the average protein-coding gene (Fig. S2 B). Given that the majority of known localization signals reside in 3′UTRs (Tushev et al., 2018), we interpret our mRNA localization results as being representative of the whole genome. Similarly, 3′UTRs extensions often contain sequences that regulate mRNA stability, suggesting that the 200 CPTI genes are likely to be similar to the rest of the genome, at least in the characteristics of their 3′UTR extensions.

To assess how representative the 200 genes in our screen are for gene function in the genome, we compared the total number of unique parent gene ontology (GO) terms (GOSlim terms) associated with the genes in our dataset to the number of unique GO terms found in all protein-coding genes. The genes in our dataset map to 89.9% of the terms across all three GO categories (the GO categories are available in Table S1), which makes the collection highly representative of the functional diversity of protein-coding genes. Together, these results indicate that we are not significantly undersampling the complexity of gene

expression patterns and that the percentage of genes with a given expression pattern in our sample could be extrapolated to provide an estimate of the total number of transcripts with that expression pattern across the whole transcriptome.

### A generalizable workflow for assembling and browsing integrated microscopy and bioinformatics databases

Extracting biological insight from large microscopy datasets is a notoriously challenging and laborious process. To facilitate analysis and browsing of our dataset, we established a generalizable workflow (Fig. 2) to display the images, annotate, and score gene expression patterns consistently across many cell types and systematically interrogate the microscopy data together with genomics data and other large scale microscopy studies. This approach makes the data easier to interpret, facilitates novel insight and hypothesis generation, and extends the functionality and utility of published resources.

The images in our dataset, like most light and electron microscopy images, contain rich and diverse 3D information that is difficult to convey in a single snapshot or a single figure in a manuscript. Each image consists of a large 3D volume in which there are multiple cell layers and multiple labels that can be used to address different biological questions throughout the volume. Moreover, each gene was characterized in multiple tissues of the nervous system, and so the combined dataset represents more than 1,000 individual figures, a data volume that cannot be published as conventional figures in a manuscript. Therefore, we developed an approach that displays selected views of the 3D image stacks simultaneously, while also providing access to the raw intensity data. Using the open source OMERO.Figure web application (Allan et al., 2012) with its links to the original data stored in OMERO (Goldberg et al., 2005). Multiple regions of interest (ROIs) from specific compartments were selected and contrasted to display specific cellular compartments from each image in an easily browsable and consistent "Figure" format, at scale.

To also analyze figures quantitatively at scale, we developed a Python application to systematically annotate OMERO.Figure images, which we named Annotate.OMERO.Fig (see Materials and methods). The scoring application takes a customizable set of questions, which are presented to scorers via a graphical user interface as it cycles through an image dataset. Then, the user-scored answers are collated and exported in a spreadsheet format for downstream analysis. Three experts annotated each tissue independently by answering the same standardized questions, such as, "Is RNA present in axon terminal?" Where

**A** **Generalizable pipeline for display and analysis of a multi-tissue gene expression database**

VNC neuropil

optic lobe neuropil

mushroom body neuropil
NB lineage (Fig. 1)

nerve fibers and glia

*Drosophila* larva fillet prepation

CNS overview

Regions of interest

neuromuscular junction (NMJ)

nuclei YFP protein *YFP* mRNA

**B** OMERO.figure

python

**1. Store images and display multiple ROIs**

**2. Annotate images and analyze expression patterns**

MDV

InterMine FlyBase

GENEONTOLOGY
Unifying Biology

| Name | Date Modified | Size | Kind |
|---|---|---|---|
| your_data.csv | 13/06/2019 | 282 KB | Comma Separated Spreadsheet (.csv) |
| RNAseq_data.txt | 13/06/2019 | 7.7 MB | Plain Text Document |
| Proteomics_data.csv | 13/06/2019 | 517 KB | Comma Separated Spreadsheet (.csv) |
| iCLIP_data.txt | 13/06/2019 | 7.7 MB | Plain Text Document |
| eCLIP_data.csv | 13/06/2019 | 15 KB | Comma Separated Spreadsheet (.csv) |

**4. Browse image and bioinformatics databases, identify novel gene targets and hypotheses**

**3. Cross-reference image analysis with data repositories and published datasets**

Figure 2. **A custom image annotation application and generalizable workflow to assemble and browse integrated imaging and bioinformatics databases. (A)** Images are obtained from multiple nervous system compartments. **(B) (1)** Microscopy data is stored on an OMERO server and adjusted for

multidimensional display using the OMERO.Figure web application. **(2)** A customizable user interface was developed in Python to display and annotate the OMERO.Figure images. **(3)** A Python application with graphical user interface is used to write annotations to a database along with queries from publicly available bioinformatics datasets. **(4)** The database can then be imported into the MDV web application to intuitively explore the data and discover hidden functional associations with machine learning algorithms. An MDV collection with our full dataset can be accessed via https://doi.org/10.5281/zenodo.6374011.

the expert annotators disagreed, we used a majority vote approach to select the correct answer (Fig. 3). Since it was important to also view the results in the context of what is already known about a gene, we added a script to extract data from specified databases, either from online repositories or directly from local files, and merge the data into a single file. This approach allowed us to browse images associated with a specific gene, while simultaneously viewing its gene ontology, relative expression levels in published transcriptomic and proteomic datasets, and genetic screens. Moreover, assembling the imaging data in its rich bioinformatic metadata context made it possible and convenient to deploy machine learning algorithms for hypothesis generation and gene candidate selection to guide future experiments.

To facilitate manual curation and data browsing, we uploaded the annotation file and associated images into a web application called MDV, an open-source software platform that enables integration of a large number of images with a rich and diverse landscape of bioinformatics datasets. We designed and built a pipeline that is easily generalizable to other model organisms and data repositories. Our image dataset includes 1,361 Figures from 200 genes in the CPTI gene trap collection, with downstream analysis of the whole genome, allowing for extrapolation of the findings to predict additional genes with similar expression patterns or phenotypes. Moreover, the dataset lists other genes with known protein trap insertions and links directly to Intermine (Smith et al., 2012), FlyBase (Larkin et al., 2021), Gene2Function (Hu et al., 2017), SFARI annotation (Banerjee-Basu and Packer, 2010), all of which extend the utility of those resources. The MDV platform allows point-and-click filtering by GO terms, disease associations, or characteristics of genes such as 3′UTR length or number of isoforms. The MDV collection this study can be viewed at https://doi.org/10.5281/zenodo.6374011.

### Overview of the screen results

A fundamental question we addressed with this dataset is where proteins and mRNAs from 200 different genes are expressed relative to each other in various intracellular compartments and between different parts of the nervous system. 97.5% of genes (195/200) show discordance between mRNA and protein expression in at least one cellular compartment in the nervous system (Fig. 3 A). The data in Fig. 3 A are grouped to distinguish intercellular discordance (between cells) from intracellular discordance (within a single cell), which are likely to arise from distinct mechanisms. Intercellular mRNA/protein discordance primarily occurs through differences in transcription, translation, or degradation rates between different cells. In contrast, intracellular mRNA/protein discordance is likely to be dependent on transport and localized translation, as well as translational repression. The two types of discordance were observed at similar frequencies in our dataset (51% intercellular discordance and 49% intracellular discordance).

To further dissect the details of expression patterns across the nervous system, we tabulated the percentage of genes expressing either mRNA, protein, both, or neither in each compartment (Fig. 3 B). This representation highlights that in most compartments discordance arises from the presence of mRNA without protein (compare light pink to dark green segments), which is highly indicative of translational repression. The major exception is synaptic neuropil in the mushroom body, optic lobe, and VNC, which all have higher percentages of genes that only express proteins. We hypothesize that this expression pattern could be established by selective protein transport to the synapse or higher protein stability that allows it to persist after the mRNA is degraded. The latter could also explain the presence of protein where there is no mRNA in the soma (Fig. 3 C). In general, fewer genes express RNA or protein in the synaptic compartments, suggesting that synaptic gene expression is highly selective. Together, these representations of the data highlight that the dataset is a unique and valuable resource for exploring discordance between mRNA and protein for each gene in each compartment at high 3D spatial resolution and very high sensitivity of single-molecule mRNA detection.

### Post-transcriptional control of neuroblast differentiation

Although transcription factors have been thought to be the primary regulators of neuronal differentiation, post-transcriptional regulation also plays a major role in nervous system biology (Cajigas et al., 2012). To assess the prevalence of post-transcriptional regulation in neuronal differentiation, we applied our mRNA and protein reporter approach to visualize gene expression in populations of asymmetrically dividing neuroblasts in the larval central brain (Fig. 4), a powerful and well-used model for understanding neural differentiation (Homem and Knoblich, 2012). Neuroblast lineages are typically grouped into a single compartment and surrounded by glial cells providing a glial niche, which can allow the neuroblast lineage to be identified unambiguously. We performed our scoring in these clearly identifiable neuroblast lineages. We discovered that post-transcriptional regulation is unexpectedly widespread among genes that are selectively expressed in neuroblast lineages (see Table S3 for detailed analysis associated with Fig. 4). Approximately one-third of the genes with cell-specific expression in our dataset show discordance between protein and mRNA expression in neuroblast lineages (Fig. 4 A), a hallmark of post-transcriptional regulation. Cell-specific expression patterns were observed for 21.5% of the genes (43 genes), whereas 57% of the genes (114 genes) were expressed homogeneously throughout the neuroblast lineage (Fig. 4 B) and 21.5% of genes (43 genes) were not detected above background at either the mRNA or protein level (Fig. 4 C). We found that every gene that expresses mRNA

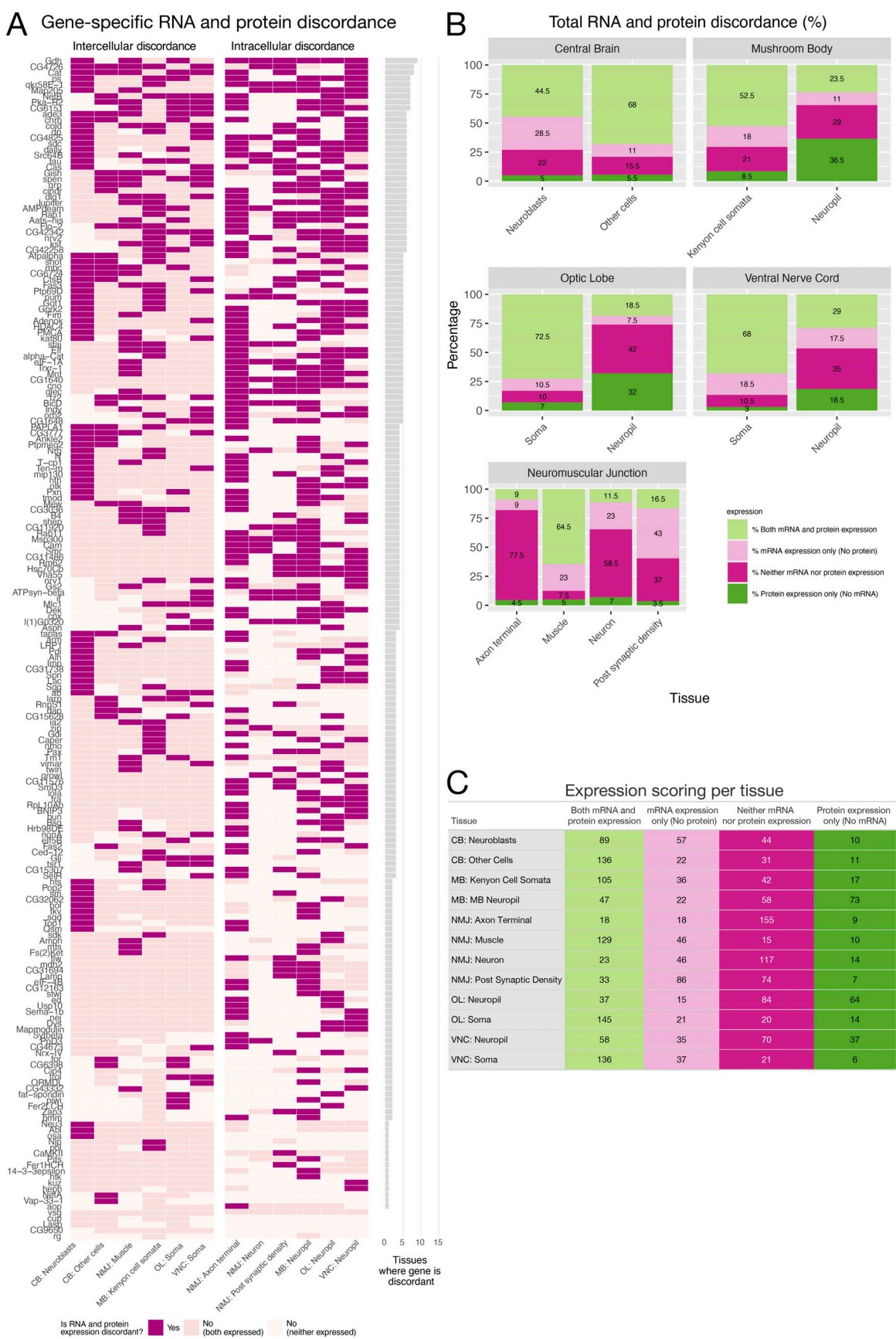

Figure 3. **Summary of annotations for mRNA and protein expression for each tissue and discordance. (A)** Discordance per gene per tissue is shown in dark purple and concordance in two shades of light pink to indicate how concordance arises. Each row corresponds to a gene and the reader can see how many

tissues show discordance per gene. **(B and C)** (B) Percentage of genes scored and (C) numbers of genes scored that show expression of mRNA and protein or the absence of expression of mRNA and protein per tissue and compartment. The graph also shows percentages of mRNA expressed and protein not expressed and vice versa per tissue and compartment. CB, central brain; MB, mushroom body; OL, optic lobe.

or protein in neuroblasts is also expressed in their immediate progeny, indicating that none of the 200 genes in this dataset are strictly neuroblast specific in their expression. In fact, most of these genes are expressed broadly throughout the central brain. However, one gene, *indy*, is highly transcribed in neuroblasts and a single ganglion mother cell before it is rapidly

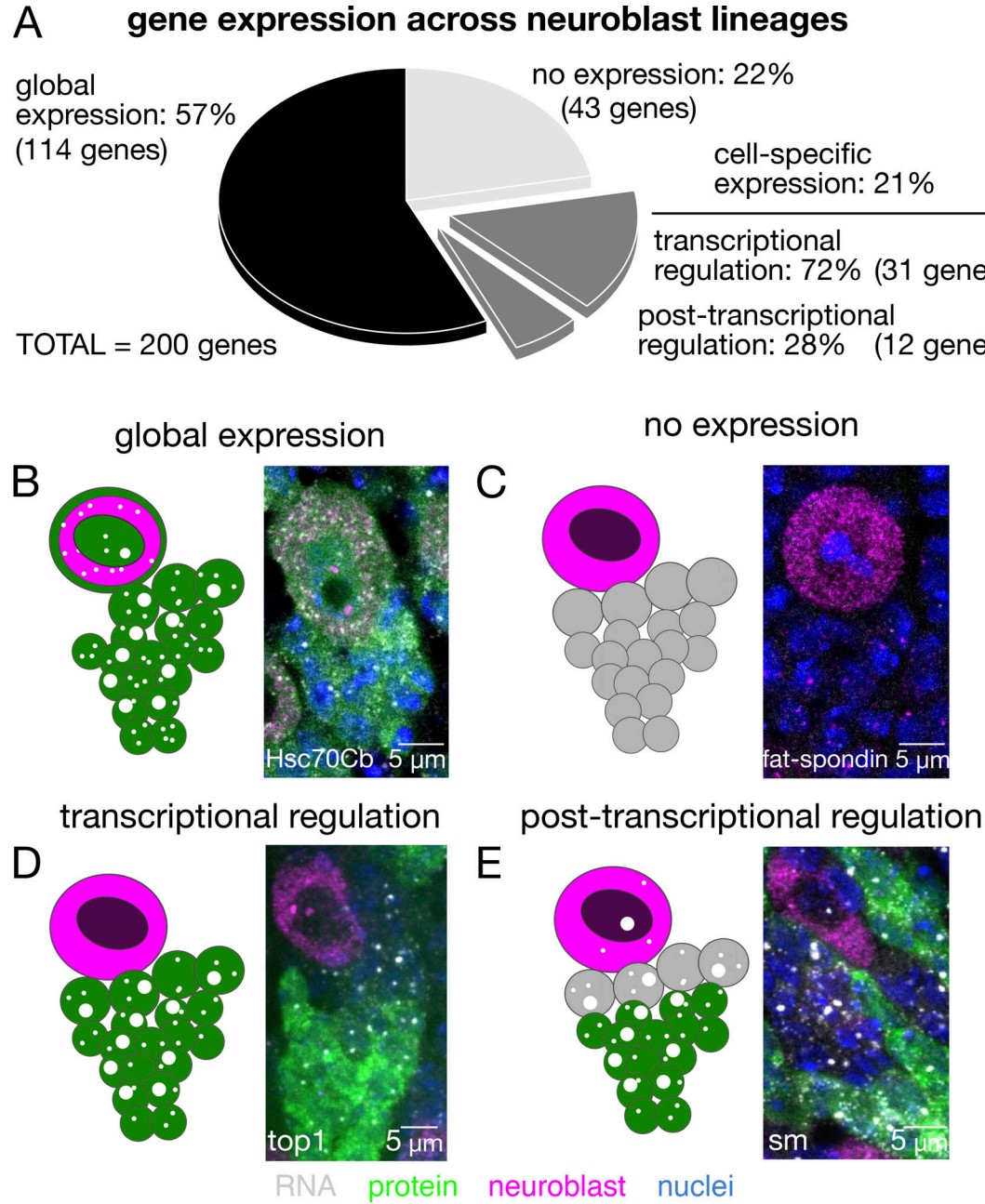

Figure 4. **Discordance between protein and mRNA expression patterns reveals the extent of post-transcriptional regulation in neuroblast differentiation.** 200 YFP-reporter lines were imaged across multiple type-I neuroblast lineages in the larval central brain. **(A)** Pie chart showing the relative distribution of different gene expression patterns. **(B and C)** The majority (57%) of genes were expressed homogeneously throughout the neuroblast lineage (B), while 22% of genes were not detected in the central brain region (C). The remaining 21% of genes exhibited some degree of cell-specific expression throughout the neuroblast lineage. **(D and E)** In some of the 21%, where protein levels were either correlated with mRNA and transcription levels indicating transcriptional regulation (D) or not correlated with mRNA levels indicating post-transcriptional regulation (E). Over one-quarter of all genes with cell-specific expression patterns exhibited this hallmark of post-transcriptional regulation.

shut off (Fig. S3). We conclude that post-transcriptional regulation is likely to play a widespread role in neuroblast biology and differentiation of its progeny.

Of the 43 genes with cell-specific expression patterns, 31 (72%) genes exhibit highly correlated protein and mRNA expression between different cell types. Of those 31 genes, only a subset of cells actively transcribes the gene, and each cell that produces mRNA also produces protein. Highly correlated protein and mRNA expression is a strong indication that these genes are transcriptionally regulated. A representative gene with this expression phenotype is *top1* (Fig. 4 D), a topoisomerase that has essential functions in cell proliferation. Of the 43 genes with cell-specific expression, 12 (28%) exhibit obvious discordance between mRNA and protein levels throughout the neuroblast lineage (Fig. 4 A). The transcription rate of these genes, as indicated by the relative intensity of smFISH nuclear transcription foci, is similar across the neuroblast lineage; however, protein signal is only detectable in a minority of the progeny cells (Fig. 4 E). Discordance between protein and mRNA content is a strong indication of post-transcriptional regulation and also suggests an important cell-specific function. Consistent with this idea, four of the 12 genes with discordant expression patterns, *pbl*, *Rm62*, *qkr58E-1*, and *cno*, were previously identified in a genome-wide screen surveying neuroblast division phenotypes (Neumüller et al., 2011).

Some cells in Fig. 4, B and E, have cytoplasmic mRNA in the absence of obvious transcription foci. The simplest biological interpretation is that transcription at those loci has been turned "off" while there are still mRNAs present in the cytosol, and pre-mRNAs are quickly exported out of the nucleus after transcription termination and splicing, therefore it is not surprising to detect cytosolic mRNAs in the absence of highly active transcription foci. It is also possible that the transcription foci were not captured in the optical section. Detection of transcription foci is actually far more robust than cytosolic mRNA detection— there are multiple pre-mRNA in a diffraction-limited area as opposed to single transcripts in the cytoplasm—so it is unlikely that transcription sites would be undetectable by smFISH if single mRNAs in the cytosol are detected.

### Synaptic mRNA localization across different central nervous system (CNS) neuropils

mRNA localization provides an additional layer of post-transcriptional regulation to target specific proteins to neural synapses. To determine the contribution of mRNA localization to synaptic proteomes, we visualized mRNA and protein content across multiple neuropil regions of the intact larval brain. We found that nearly half of the genes in our dataset express proteins that are relatively abundant at mushroom body synaptic regions, at the periphery of cells. Nearly one-third of the genes also express mRNA that is present at the synapse. However, the sets of synaptic mRNAs and synaptic proteins do not overlap entirely, providing insight into the specific mechanisms of localized post-transcriptional regulation (see Table S2 for detailed analysis).

To analyze these localization patterns further, we acquired stacks of confocal images from three synaptic regions in the larval nervous system, the mushroom body, the optic lobe neuropil, and the sensorimotor neuropil of the VNC. For each of the 200 randomly selected YFP protein trap lines, we assessed whether the protein and/or mRNA was expressed in soma or synaptic neuropil for each ROI. In the mushroom body, 94 out of the 200 genes in our dataset (47%) encode proteins that are detectable either in the mushroom body calyx or one of the lobes (Fig. 5, A–H). Of those 94 genes, 30 (32%) are encoded by mRNAs that are also detected in either region of the mushroom body (Fig. 5, A–D). These observations suggest that localized mRNA translation contributes about one-third of the synaptic proteome, slightly less than what has been previously reported (Zappulo et al., 2017). Surprisingly, another 59 transcripts are present at synapses without detectable levels of protein (Fig. 5, I–L), and unexpectedly, many of those genes encode proteins with predominantly nuclear localization and function (Fig. 5, I–L, and Q). We hypothesize that these mRNAs are translationally silenced and primed to produce proteins that will transduce synaptic input to the nucleus. However, we cannot exclude the possibility that these mRNAs encode proteins that are present at levels below our detection limit or that these transcripts are present in neuropil without performing a function.

We reasoned that it would be unlikely for synaptic localization of mRNA that lacks any function to occur consistently across different cell types. Therefore, we repeated the localization analyses in different types of neurons, including the optic lobe and VNC neuropils. We found that 28 of the 67 (42%) mRNAs present at the mushroom body synaptic neuropil are also present at the optic lobe neuropil (Fig. 5 R). Moreover, many of the synaptic mRNAs that encode nuclear proteins were also present in the optic lobe neuropil (Fig. 5, M–P, and R).

To gain further insight into which molecular functions are cell specific or common across all three synaptic compartments, we performed GO enrichment analysis of genes with discordant RNA and protein expression (Fig. S4). Discordant expression across all synapses were terms that are obviously related to mRNA localization and asymmetric function, such as cell–cell junction assembly, apical cytoplasm, and cell periphery. Surprisingly, the genes with discordant expression in specific compartments were mostly enriched for unique functional terms related to development. This suggests that local expression of a common set of genes supports synaptic function while a cell-specific repertoire of local transcripts guides synapse development.

### mRNA and protein localization in glia

Like neurons, many glial cell types have long and elaborate filamentous processes that are likely to require localized gene expression control through mRNA localization and local translation. Though localization of mRNAs encoding glial fibrillary acidic protein and myelin basic protein have been extensively characterized (Medrano and Steward 2001; Müller et al., 2013), relatively little is known about the hundreds of other localized mRNA species that have recently been identified in mammalian glial processes, and almost nothing is known about mRNA localization in *Drosophila* glia.

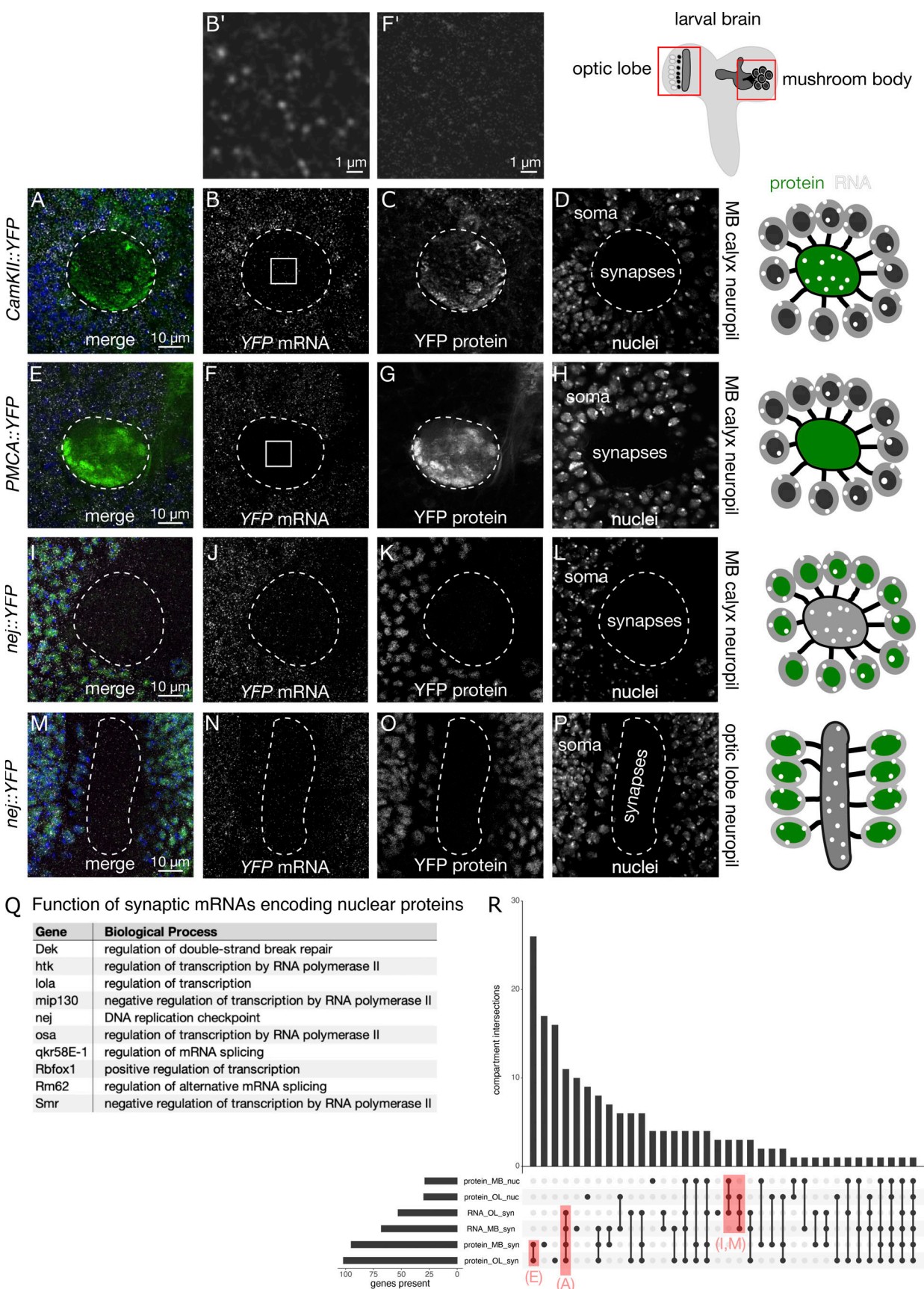

Figure 5. **Discordance between synaptic localization of mRNA, protein, and synaptic function is surprisingly prevalent. (A–H)** Optical sections of the larval brain showing mRNA and protein distribution patterns in the mushroom body (MB) and optic lobe (OL) neuropils. 32% of synaptic proteins have mRNAs

at the synapse (A–D), while 68% of synaptic proteins are expressed without localized mRNAs (E–H). **(B′ and F′)** High magnification images of the neuropil regions outlined by a white box in panels B and F. **(I–Q)** Nearly half of the synaptic mRNAs fail to generate detectable levels of synaptic protein; instead, these mRNAs tend to encode nuclear proteins in both the MB (I–L) and OL (M–P) neuropils. Surprisingly, many of the proteins encoded by synaptic mRNAs are transcription factors (Q). **(R)** UpSet plot showing the number of genes expressed in each compartment, at the mRNA and protein level, for the entire dataset.

To identify new genes with mRNA localized to glial processes in the larval CNS, we manually searched the image dataset for protein and mRNA expression patterns that show enrichment in glial processes. There are six types of glia that have invariant positions and characteristic morphologies within the nervous system (Schmidt et al., 2012). We focused on cortical glia in the central brain region and ensheathing glia that envelops the mushroom body neuropil, as the processes in these cells are easily identifiable. For reference, we labeled those regions by expressing membrane-bound mCherry specifically in glial cells using the *repo-GAL4* driver (Fig, 6, A, B, and E).

To assess glial localization for the 200 genes of interest, we used a pan-glial GAL4 driving a membrane mCherry marker (*repo-GAL4>UAS-mcd8-mCherry*) to learn the expression pattern of all glial cells, and then classified the pattern in the YFP lines (without the marker) based on knowledge of that expression pattern. To validate that the *repo-GAL4>UAS-mcd8-mCherry* indeed faithfully represented the location of glial cells throughout the nervous system, we assessed the overlap of the *repo-GAL4>UAS-mcd8-mCherry* labeling with the *Nrv2::YFP* insertion, one of the 200 lines which was already known to be a wrapping glial marker (Yadav et al., 2019). We performed smFISH using probes against the YFP sequence and found that *Nrv2* protein and mRNA are highly expressed in glial processes of both cell types (Fig. 6, C, D, F, and G), demonstrating that we can accurately classify glial mRNA localization based on the stereotypical glial morphology observed in the protein expression pattern.

Focusing on cortical and ensheathing glia in the CNS, we found that 19.5% of the proteins in our dataset are expressed in cortex glial processes in the central brain and only 11.5% of proteins are expressed in mushroom body neuropil glial processes. A very high percentage of mRNAs that encode those proteins are also localized to glial processes, 92 and 65% for the cortex and neuropil glia, respectively (Table S3).

We extended our analysis of glial mRNA localization to glia in peripheral nerve fibers and at the neuromuscular synapse. Perineural and subperineural glia wrap the outer layers of the nerve bundle, wrapping glia envelope single axon fibers. The glia also form the blood–nerve barrier between the axon and extracellular fluid. Perineural and subperineural processes extend into the neuromuscular synapse. Each of these cell types is marked by the *repo > mCherry* reporter along the fiber (Fig. 6, H and I) and at the neuromuscular synapse (Fig. 6, L–O). *Nrv2* mRNA and protein are distributed throughout each layer of glia in both the nerve fiber (Fig. 6, J and K) and axon terminals (Fig. 6, N and O) with highest expression in the wrapping glia, which form the blood–nerve barrier between the axon and extracellular fluid. We also show that mRNAs, e.g., *gs2*, are localized in the subperineural and perineural glia that are associated with more distal boutons (Fig. S5). Focusing on axon terminals at the larval NMJ, where glial protein expression patterns are

easily identifiable, we found 19 genes (9.5%) with protein or mRNA expression in either wrapping glia at axon terminal, or perineural and subperineural glia of the NMJ, and 95% of those genes also showed mRNA localization in the glial processes (Table S3). Together, analysis of mRNA and protein expression in glial processes of the CNS and peripheral nervous system shows potential contribution of mRNA localization to the proteome in that compartment (Table S3; von Kügelgen and Chekulaeva, 2020; Giandomenico et al., 2022; Zappulo et al., 2017). Our results present the first examples of mRNA localization to glial processes in the larval CNS, segmental nerve, and NMJ, highlighting a hitherto unrecognized important general function for mRNA localization and most likely, localized translation.

## mRNA and protein localization at neuromuscular synapses

mRNA localization to motor axon terminals is one of the most extreme examples of polarized gene expression in metazoans, with mRNAs being transported, within some neurons, a distance nearly equivalent to the entire body length of the animal. Neuromuscular synapses on the larval body wall muscles provide an excellent system to investigate such long-distance axonal mRNA transport, to determine whether the mRNAs are pre- or postsynaptic, a question that is much harder to address in the dense synaptic neuropils of the central brain. To investigate how frequently different mRNAs are localized, we applied our mRNA and protein trap microscopy screening approach to the larval NMJ. We found that the presence of mRNA in these motor axon terminals is relatively rare, as is strong enrichment of mRNA in the postsynaptic density (PSD) in the muscle cells.

We combined our smFISH and protein trap approach with subcellular markers to distinguish individual axon terminals, the PSD, and muscle nuclei. We found that 13.5% of the genes in our dataset encode proteins that are detectable in the motor axon terminal (Table S2). Around two-thirds of those proteins are accompanied in the axon terminal by the mRNAs that encode them (Table S2), which is consistent with the percentage of transcripts that are typically detected in transcriptome-wide studies (von Kügelgen and Chekulaeva, 2020). An example of a gene with this expression pattern is *sgg* (Fig. 7, A–D), which we chose to further characterize because of its known role in NMJ development (Fig. 8). The NMJ axon terminal compartment contained far fewer mRNA species than mushroom body, optic lobe, and VNC neuropils. These results suggest at least one of three plausible mechanistic explanations, none of which are mutually exclusive. One is that the motor axons have a different repertoire of RNA binding proteins restricting entry into the axon (Martínez et al., 2019) and/or transport to the axon terminals. Another possible explanation is that only the most stable mRNAs are able to avoid degradation across the extremely long distance from the soma to the NMJ. Finally, transcripts that are

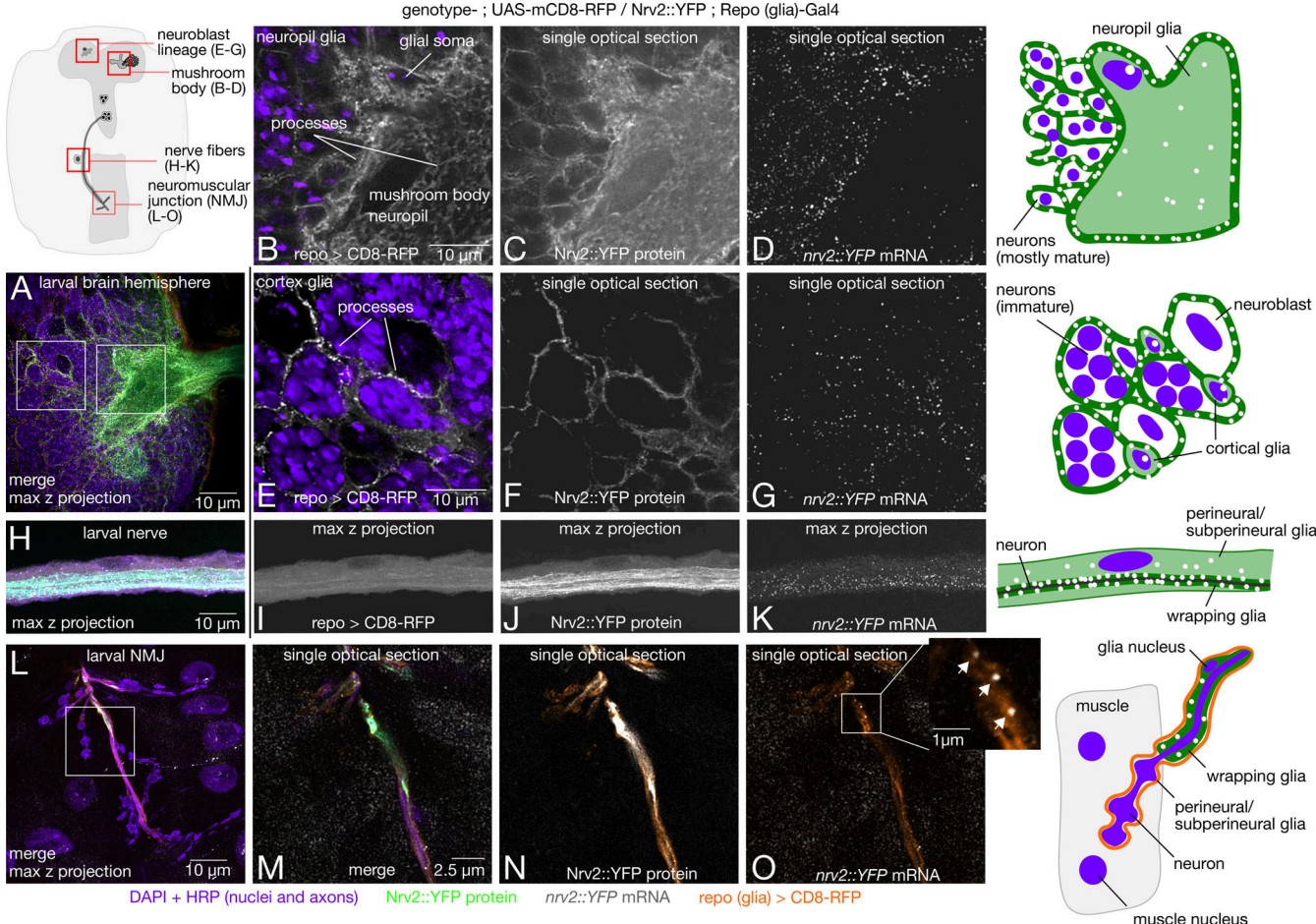

genotype- ; UAS-mCD8-RFP / Nrv2::YFP ; Repo (glia)-Gal4

DAPI + HRP (nuclei and axons)    Nrv2::YFP protein    *nrv2::YFP* mRNA    repo (glia) > CD8-RFP

Figure 6.    **mRNA is localized in glial processes throughout the larval nervous system.** *Repo (glia)-GAL4* and *UAS-mCD8-mCherry* (membrane marker) were crossed into the Nrv2::YFP background to demonstrate *nrv2* mRNA (gray) and protein (green) localization in glial processes (orange) throughout the larval nervous system. **(A)** Overview confocal maximum intensity projection image of a larval brain hemisphere showing the relative locations of glial cells. **(B–G)** Single optical sections showing spatial overlap of *nrv2::YFP* RNA and protein channels with the *repo > mCherry* marker in neuropil (B–D) and cortical (E–G) glia. **(H–K)** Representative image of *nrv2::YFP* mRNA and protein expression in segmental nerves innervating the larval body wall musculature. **(L)** Overview of glia anatomy and *nrv2::YFP* expression at the larval NMJ. **(M–O)** Single optical sections showing spatial overlap between glial processes, Nrv2::YFP protein, and single mRNA molecules (inset, arrows).

detected in the motor axon terminals could have distinct localization signals.

To address the degree to which mRNA localization is likely to contribute to targeting proteins in the postsynapse, we characterized in detail the distribution of the 200 mRNAs and the proteins they encode at the muscle and postsynaptic cytoplasm. Our dataset shows that a large proportion of genes encode mRNAs that are present, but not enriched, within the PSD without any corresponding protein enrichment or known synaptic function for the proteins encoded by the mRNAs (Table S3). For example, the distribution of *nrv1* and *zap3* mRNAs are indistinguishable, even though the Nrv1 protein is strongly enriched at the PSD and Zap3 protein is evenly distributed throughout the muscle cell (Fig. 7, E–L). We identified 13 in total with strong enrichment at the PSD, none of which show obvious mRNA enrichment in the same location (Table S3). Nevertheless, mRNA from a large proportion of genes is present at the PSD, despite not being enriched in that compartment relative to the rest of the muscle cytoplasm. We interpret these results as

indicating that mRNA localization does not play as strong a role in the NMJ postsynapse. Given that in the postsynaptic (muscle) compartment, translation factors, such as eIF4E, are known to regulate NMJ development and plasticity (Menon et al., 2004), we propose that spatial regulation of gene expression makes a strong contribution to the postsynapse but through localized translation, as in the case of Msp300 (Titlow et al., 2020).

### Active *sgg* mRNA transport and activity-dependent accumulation of *sgg* in ghost boutons

Localization of mRNA in the *Drosophila* NMJ axon terminal has not previously been demonstrated at the single-molecule level. To determine whether *sgg* mRNA is actively regulated during plasticity at the larval NMJ, we performed a set of experiments using transport mutants and a spaced KCl stimulation paradigm (Ataman et al., 2008). We found that not only is *sgg* actively transported to the axon terminal, Sgg protein levels in the axon terminal are elevated in response to KCl stimulation, and both *sgg* mRNA and protein appear in newly formed synaptic

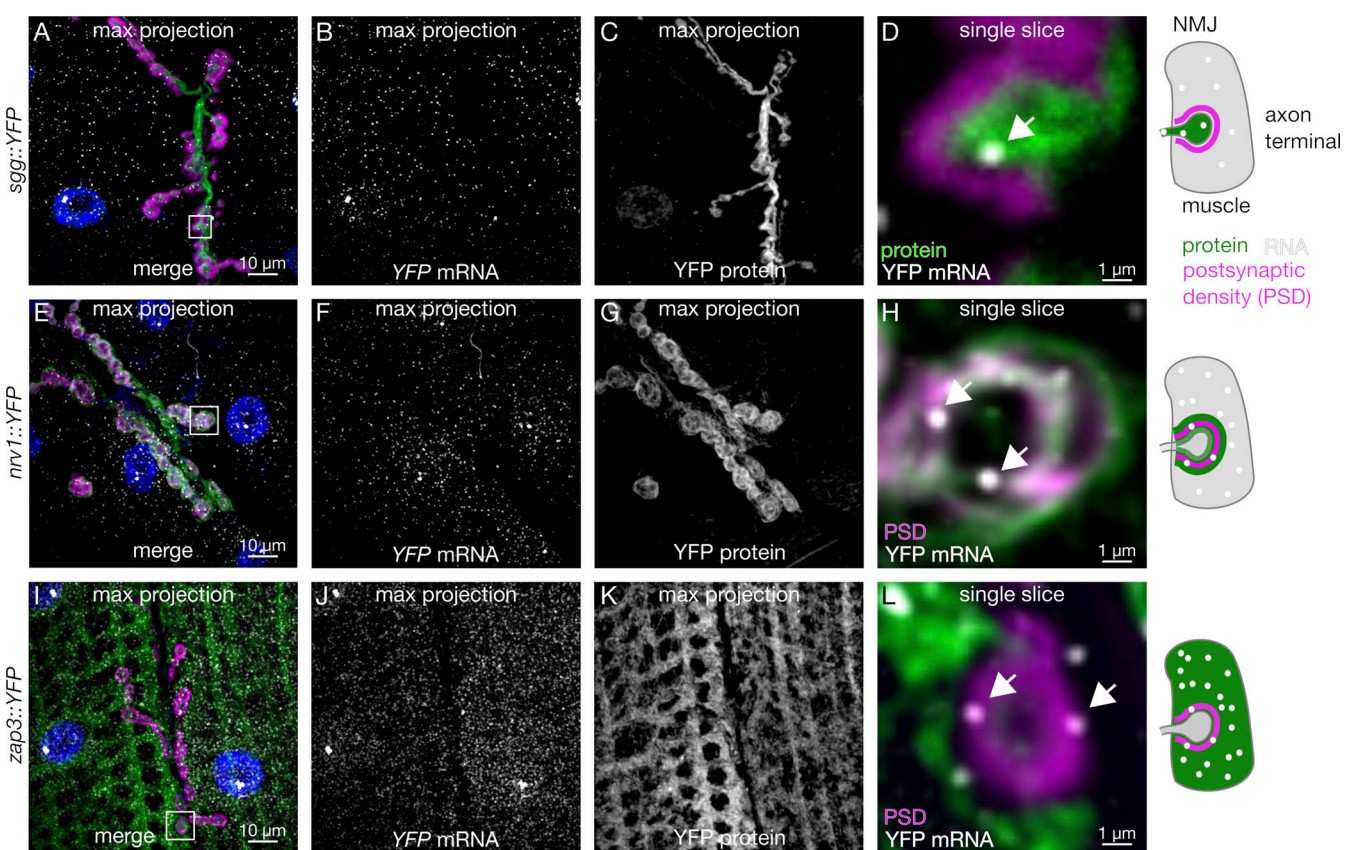

Figure 7. **mRNA is present on both sides of the larval neuromuscular synapse. (A–C)** Maximum intensity projection of confocal images showing *sgg::YFP* mRNA and protein localization at the larval NMJ. **(D)** Single optical section of the region in A (white box, 10× magnification) showing protein and individual mRNA molecules (arrow) located in the axon terminal. **(E–G and I–K)** Max projections of *nrv1::YFP* and *zap3::YFP* expression, which have very distinct protein expression patterns despite nearly indistinguishable mRNA patterns. **(H and L)** High magnification single optical sections show that mRNA (arrows) from both genes is present within the PSD.

boutons. We anticipate that at least a number of other transcripts we identified in the motoneuron axonal synapses will be similarly locally translated in response to neuronal activation. However, carrying out such experiments systematically for 69 genes (Table S2) is beyond the scope of this study. Nevertheless, our demonstration of local translation of Sgg protein raises the possibility that many more low-abundance mRNAs at the distal axonal synapses are locally translated.

Kinesin-1 is known to be required for transport in neurons in many circumstances and tissue types. To determine the mechanism for *sgg* mRNA localization to axon terminals of the larval NMJ, we carried out smFISH experiments on *kinesin1 heavy chain* mutant third-instar larval NMJs. We found that *sgg* mRNA is actively transported to motor axon terminals by the kinesin-1 motor. The number of *sgg* mRNA molecules in axon terminals was measured by counting the number of diffraction-limited fluorescent spots in the images of smFISH within a 3D-segmented axonal volume. *sgg* mRNA measurements were acquired in wild-type larvae, and in a trans-heterozygous *khc* mutant (*khc²³/khc²⁷*), a combination of an amorphic and a hypomorphic allele that avoids lethality. Loss of *khc* function resulted in an 84% reduction (Fig. 8 E) in the number of *sgg* transcripts per NMJ (Fig. 8, A–C). We asked if the absence of khc transport causes sgg transcripts to accumulate in the motor

neuron soma; however, there was no evidence of increased sgg in khc²³/khc²⁷ mutant VNCs (Fig. S6). While it is important to consider that the mRNA localization phenotype occurs in the context of abnormal synaptic and organism development in the trans-heterozygous *khc* mutant background (Gardiol and St Johnston, 2014; Kang et al., 2014), the result strongly indicates that kinesin-1–based transport is required for *sgg* localization at the NMJ.

We then asked whether kinesin-1–based transport is required for activity-dependent synaptic plasticity using a patterned chemical stimulation assay that induces the formation of new synaptic boutons, a form of activity-dependent synaptic plasticity that requires protein synthesis (Ataman et al., 2008). In this assay, boutons with presynaptic labeling that have not yet acquired the PSD marker Dlg1 are immature, so-called "ghost boutons." Using KCl stimulation and ghost bouton labeling as a readout of structural plasticity, we found that loss of *khc* function resulted in an 85% reduction in the number of activity-induced ghost boutons (Fig. 8, F–H). Although loss of *khc* function also disrupts protein and organelle transport, these experiments show that *sgg* mRNA is actively transported to the synapse and that kinesin-1–based transport is generally required for structural plasticity at the larval NMJ.

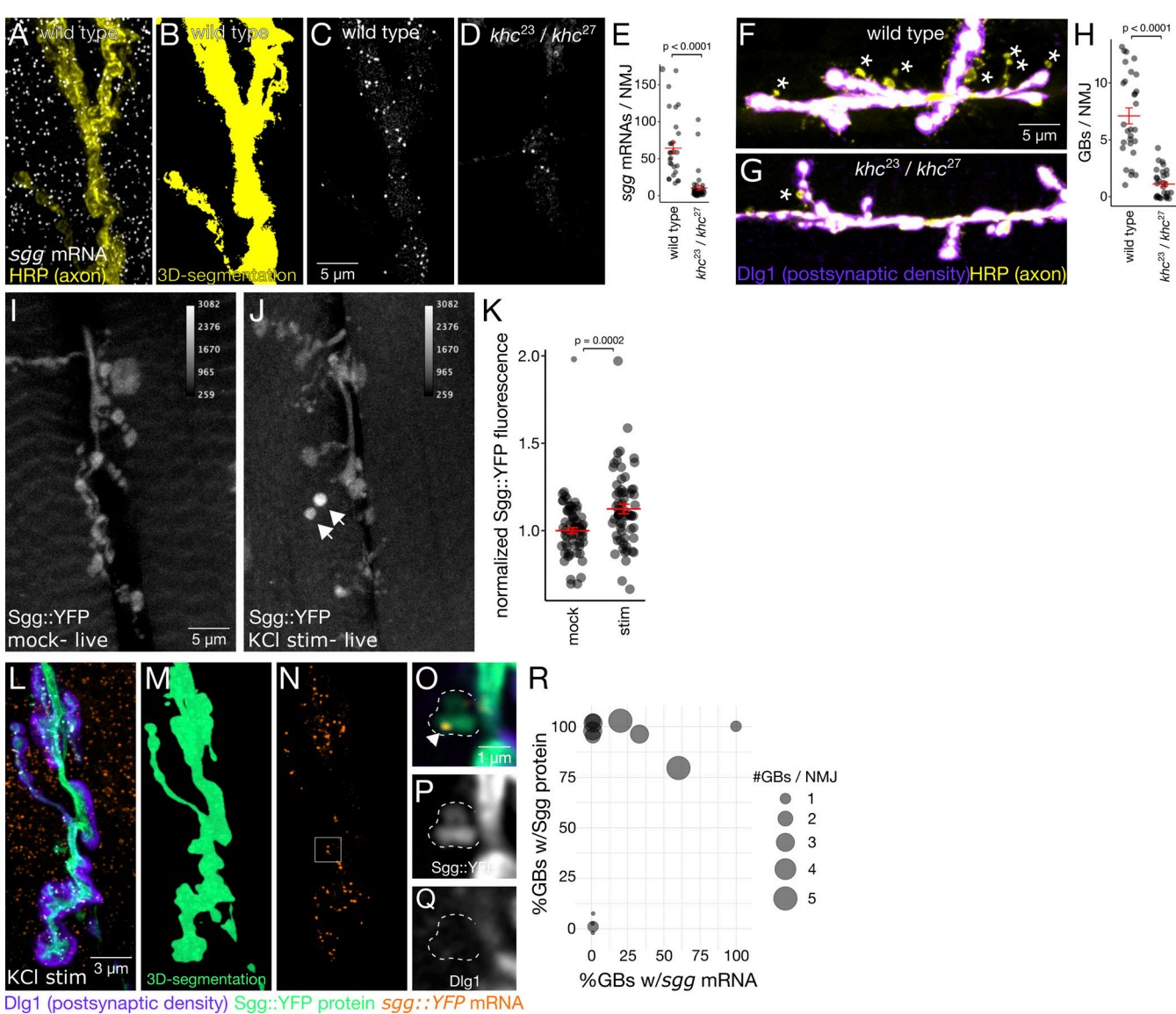

Figure 8. ***sgg* mRNA is actively regulated at the larval NMJ. (A–E)** *sgg* mRNA localization at axon terminals requires kinesin. **(A–D)** Maximum z-projection image showing *sgg* smFISH signal at the axon terminal (yellow; A). Images were segmented in 3D with the axon marker channel (B), revealing a significant decrease in the number of axonal *sgg* mRNAs in *khc²³/khc²⁷* transheterozygous mutants (C and D). **(E)** Quantification of *sgg* transcript levels in axon terminals (mean ± SEM; one-tailed Student's *t* test). **(F–H)** Kinesin is required for ghost bouton (GB) formation. Max z-projection images show GBs (asterisks) in KCl stimulated NMJs from kinesin mutants and wild-type controls. **(H)** Quantification shows significantly fewer GBs in kinesin mutants (mean ± SEM; one-tailed Student's *t* test). **(I–K)** Live imaging of axon terminals shows a significant increase (mean ± SEM; one-tailed Student's *t* test) in Sgg protein levels in samples stimulated with five pulses of KCl. Arrows indicate activity-induced GBs. **(L–R)** In fixed samples, Sgg protein and mRNA are present in a large percentage of GBs. Axon terminals were segmented in 3D to isolate signal in the axon from signal in the muscle (M). **(O–Q)** Single optical sections show a distinct puncta (arrowhead) and Sgg protein signal in a bouton that hasn't yet formed a PSD. These sections are zoomed-in views of the region of interest indicated in *sgg* mRNA panel (N). **(R)** Bubble plot representing each NMJ with the percentage of GBs containing mRNA and protein on the X and Y axes, respectively. Nearly 100% of GBs have Sgg protein with area of the circle proportional to the number of GBs per NMJ.

The canonical function of localized mRNA is to provide an immediate source of new protein translation in response to an external stimulus. To determine whether Sgg protein levels are elevated in response to patterned KCl stimulation, we quantified Sgg::YFP fluorescence levels in KCl-stimulated larval fillet preparations relative to mock-treated controls. We found a modest but highly significant increase in Sgg::YFP levels at stimulated NMJs (12.4 ± 0.02%; P = 0.0002, Student's *t* test; Fig. 8, I–K). It is perhaps not surprising to observe such a modest

increase in protein level, given that the fluorescence intensity measurements were averaged across the entire axon terminal, while the response is expected to be highly localized. In fact, we often observe high signal intensity concentrated in individual boutons that are immature in appearance (Fig. 8 J, arrows), suggesting that Sgg protein is localized at higher concentrations in axon terminals during the early stages of synapse formation.

To determine if Sgg appears in newly formed synaptic boutons, we repeated the KCl stimulation experiment and imaged

fixed samples labeled with presynaptic and postsynaptic markers. Sgg protein was present in 93% of ghost boutons (out of 27 ghost boutons from 14 NMJs in five different animals; Fig. 8, L–R), indicating that Sgg is almost always present during the early stages of activity-dependent synapse formation. To ask whether localization of *sgg* mRNA could play a role in the accumulation of Sgg protein in ghost boutons, these samples were also labeled with smFISH probes targeting *sgg* mRNA. We detected *sgg* mRNA in over 20% of ghost boutons (Fig. 8, L–R). We interpret our results as indicating that only some of the Sgg protein is translated from mRNA locally, whereas some Sgg protein is likely to be transported to the synapses in response to activation.

## Discussion

We present a data resource and a generalizable strategy to investigate the mechanisms of spatial gene expression control for a large number of gene candidates at subcellular resolution and across multiple whole tissues. To facilitate the extraction of new biological knowledge from this dataset, we have developed a computational pipeline to annotate and browse the image data and systematically interrogate the imaging data alongside existing genomic and phenotypic datasets. This approach has yielded insight into post-transcriptional regulation that improves our understanding of both brain development and synaptic plasticity.

### A powerful method to quantify the entire gene expression life cycle for any gene

Gene expression is a multistep process that is rarely investigated end-to-end, from transcription to mRNA processing to protein production. Our approach to measuring gene expression provides important insight into how an individual gene is regulated in vivo, while also highlighting the need to understand mechanisms of post-transcriptional regulation in more detail. This approach could be especially powerful in model organisms where large collections of protein traps are already available and for tissue types in addition to the nervous system in *Drosophila*. We developed software tools that make it easier to systematically assemble, annotate, and classify the imaging data for curation (Fig. 2).

### Estimating the contributions of post-transcriptional regulation to brain development

Our dataset highlights a set of genes that exhibit obvious discordance between mRNA and protein expression levels throughout the neuroblast lineage in the larval central brain. This result is consistent with bulk sequencing studies that have identified large sets of genes that have mismatched levels of mRNA and protein (Liu et al., 2016; Buccitelli and Selbach, 2020). We show, with high spatial resolution in a tissue-specific context, that lack of correlation between mRNA and protein levels often arises between cells at different stages of neuronal differentiation. Lack of correlation between mRNA and protein concentration in a cell could arise through many different mechanisms, which can be divided into two classes. First, mechanisms depending on the spatially distinct production of

protein, and second, mechanisms depending on differences in mRNA or protein decay. The quantitative power of our smFISH data can provide an estimate of mRNA synthesis and decay rates. Specifically, differences in the ratio of protein to mRNA provides an estimate of translation rates, and protein decay can be assumed to account for instances that are not explained by differences in mRNA metabolism or translation. By definition, each gene that we define as being post-transcriptionally regulated has equal levels of mRNA synthesis across the cell lineage.

The occurrence of translational regulation highlights a major gap in our understanding of translational control because it is not clear from our data whether these genes are more translationally active and/or translationally repressed across different stages of cell differentiation. Identifying the association between ribosomes and the trans-acting factors of specific mRNAs at different stages of development will be necessary to fully understand how these genes are regulated to influence neural differentiation and synaptic transmission (Halstead et al., 2016; Richer et al., 2021 *Preprint*). In addition, alternative splicing and alternative polyadenylation may influence the fate of mRNA transport and translation in specific cases, which would not be detected with our method (Tian and Manley, 2017). However, such studies can only be carried out for specific cases, and a global analysis of the trans-acting factors and their signals for the 200 genes we have characterized is beyond the scope of this study. Another point to consider is that secreted proteins could appear to be localized at a target cell where the mRNA is not expressed. This phenomenon was not observed in the neuroblast lineages in the current study but could not be ruled out for dense synaptic regions.

Our analysis revealed 12 out of 200 genes that showed obvious post-transcriptional regulation within the neuroblasts of the larval central brain (Fig. 4 and Table S3). An approximate extrapolation of that percentage to the whole genome indicates that over 500 genes are likely to exhibit similar expression patterns and post-transcriptional regulation in neuroblasts. The stability and translation of such mRNAs are known to be regulated by mRNA binding proteins. Syncrip and Imp are mRNA binding proteins that are already known to play a major role in *Drosophila* brain development (Liu et al., 2015; Samuels et al., 2020a; Samuels et al., 2020b) with conserved mechanisms in the mammalian brain. However, Syp and Imp are certainly not unique since out of the 523 known canonical mRNA binding proteins in *Drosophila* (Sysoev et al., 2016), 226 are expressed in neuroblasts (Berger et al., 2012). Characterizing the function of so many mRNA-binding proteins in neuroblast differentiation is daunting. Nevertheless, in the future, high throughput approaches may have to be brought to bear on these large numbers of genes to understand the full complexity of the more than 500 genes estimated above to be likely to exhibit post-transcriptional regulation during brain development.

### The landscape of synaptic mRNA and protein localization in an intact brain

Our study provides insight into the prevalence of synaptic mRNA localization and local translation, which is thought to be a critical factor in synapse development (Shigeoka et al., 2016; Cioni et al., 2018) and plasticity (Holt and Schuman, 2013; Holt

et al., 2019). In a related study, we demonstrated that our approach can also be applied to the adult fly brain, where we characterize protein and mRNA expression of CamKII and five other well-known synaptic genes in a specific mushroom body output neuron (Mitchell et al., 2021). Across other animal models, a series of elegant transcriptomic studies have revealed thousands of different mRNA species that are present in neurites—axons or dendrites—of various different neuronal cell types. A subset of those localized mRNAs make up a core set of neurite-enriched transcripts (von Kügelgen and Chekulaeva, 2020), and localized mRNAs are likely to encode as much as half of the synaptic proteome in cultured neurons derived from mouse embryonic stem cells (Zappulo et al., 2017). We find that a slightly lower, but similar proportion of synaptic proteins are found alongside the mRNAs that encode them in the optic lobe, mushroom body, and sensorimotor neuropils of the *Drosophila* larval brain. On average, mRNAs detected in these *Drosophila* neuropils have mammalian orthologs that localize in at least eight other synaptic transcriptome studies. Two of those genes, *ATPsynbeta* and *14-3-3epsilon*, are among the 10 most commonly detected mRNAs across the set of neurite transcriptome studies. The fact that these specific mRNAs are selectively localized to the neurite compartment, in different cell types and across millions of years of evolution, argues for the importance of their local translation in synaptic physiology.

Counterintuitively, mRNAs that encode nuclear proteins were highly enriched among the synaptic mRNAs in our dataset (Fig. 5, I–O). Retrograde signaling from synapse to nucleus is a relatively understudied process that contributes to many phases of the synaptic life cycle, including development, plasticity, and response to injury (Cohen and Greenberg, 2008; Fainzilber et al., 2011). Some of the signaling cascades that activate and execute synapto-nuclear signaling have been defined, as well as the transcription factors and genes that are upregulated in response to retrograde signaling. Consistent with synaptic transcriptome studies (von Kügelgen and Chekulaeva, 2020), we detected several mRNAs at the synapse that encode transcription factors. These transcription factors could be translated in response to local changes in synaptic activity and trafficked back to the nucleus to induce the expression of long-term memory genes. mRNAs encoding splicing factors Rm62 and qkr58E-1 were also detected at *Drosophila* synapses. Activity-dependent alternative splicing, like retrograde synapto-nuclear signaling, is poorly understood but known to be important for several phases of the synaptic life cycle (Flavell et al., 2008; Hermey et al., 2017). Our screen included some genes such as Rm62 and qkr58E-1, with functional connections to the spliceosome or splicing. The mammalian homologs of these two genes, Ddx17 and Khdrbs3, respectively, have mRNAs that are detected in at least nine synaptic transcriptome studies (von Kügelgen and Chekulaeva, 2020). Therefore, our data highlight the possibility that local synthesis and nuclear transport of Rm62 and qkr58E-1 could link elevated synaptic activity and the spliceosome.

## RNA localization in glia and the NMJ

We provide the first evidence for mRNA localization in the peripheral processes of *Drosophila* glia (Fig. 6). Like neurons, glia have long cellular processes that exhibit mRNA localization and local protein synthesis (Pilaz et al., 2016; Sakers et al., 2017); however, the functional role of mRNA localization in glia is not well defined. Our results show that several cell junction and membrane proteins are encoded by mRNAs that are localized in glial processes at the NMJ and in the CNS, cells that are functionally homologous to vertebrate Schwann cells, and oligodendrocytes, respectively. Consistent with glial transcriptomic studies, which tend to show lower transcript diversity than neurite transcriptomes, we find that the relative number of genes expressed in glial processes is lower than in neural processes. We also find that the majority of proteins expressed in glial processes have localized mRNAs (88% across peripheral and nervous system glia [Table S3]), whereas the estimated contribution of local mRNA to the synaptic proteome in neurons is ∼50% (Zappulo et al., 2017). Together, these results suggest that mRNA localization in glial processes is highly regulated and is likely to make an important contribution to the local proteome.

The extent to which glial processes influence synaptic transmission and activity-dependent plasticity at the larval NMJ is not yet known, but glial signaling through the Wnt pathway (Kerr et al., 2014) and Endostatin pathway (Wang et al., 2020) has been shown to disrupt synaptic physiology. Those signaling factors, in addition to the genes identified in our dataset, could be regulated by local translation at the neuromuscular synapse in response to extracellular cues. Moreover, the presence of localized mRNAs in peripheral processes of cortical and ensheathing glia suggest that such mechanisms could be important for cognitive function and brain development.

We were surprised by the absence of mRNA enrichment at the PSD of the larval NMJ (Fig. 7). Although 13 genes encode proteins that are highly enriched at the PSD (Table S3), none display a corresponding enrichment of mRNA. Despite the lack of mRNA enrichment, we observed a high abundance of mRNAs at the PSD encoding many different types of proteins, an indication that local translation does occur. The data are consistent with a model where specificity of local translation, for example, in response to elevated synaptic activity, is achieved through translational regulation by selective mRNA binding proteins, as shown previously for activity-dependent regulation of Msp300 by a heterogeneous nuclear ribonucleoprotein called Syncrip (Titlow et al., 2020). Here, we report another conserved mRNA binding protein, RnpS1, which has the potential to provide highly localized regulation of mRNA dynamics based on its strong enrichment in discrete punctate particles at the PSD (Fig. S7).

Given the high abundance of mRNA in the muscle cytoplasm and in close proximity to the synapse, what would be the benefit of localized mRNA? There are at least two probable functions: one is rapid and local production of protein in response to synaptic activity. It is not known how long it takes for proteins to translocate across the subsynaptic reticulum, therefore protein synthesis directly within the subsynaptic reticulum may be required to produce highly localized signaling molecules on the appropriate timescale. Another potential function is the production of PSD-specific post-translational modification (PTM) signatures. Though it has been shown that PTMs are required

for some proteins to be inserted into the PSD and form complexes with other scaffolding proteins, it is not currently known where the PTMs occur. Translation specifically within the PSD could enable compartment-specific regulation of PTMs.

Our approach provides a framework for future studies aimed at understanding gene expression control, at scale, across 3D tissue landscapes with heterogeneous cell types. By surveying a small percentage of protein-coding genes, we undoubtedly underestimate rare expression patterns, but this diverse sample provides a useful estimate of the frequency in which the observed gene expression patterns are expected to occur genome-wide. Future studies can be expanded with the use of high throughput slide scanning microscopes and the extensive collection of protein trap lines available in *Drosophila* (Nagarkar-Jaiswal et al., 2015). Additional cellular markers, for example axon and dendrite-specific markers in synaptic neuropils, and additional developmental time points would also be valuable for identifying temporal changes in mRNA dynamics. This approach to investigating gene expression provides critical insight into how gene function is regulated within the tissue environment.

## Materials and Methods
### Animal model
*Drosophila melanogaster* stocks were maintained with standard cornmeal food at 25°C on 12-h light–dark cycles unless otherwise specified. Wandering third-instar larvae were used for all experiments. The following genotypes were used: *Canton S* (wild type unless otherwise specified), *repo-GAL4* (Sepp et al., 2001), and *UAS-mCD8-mCherry* (#27391; Bloomington Drosophila Stock Center). YFP insertion lines were from the Cambridge Protein Trap Insertion project (Lowe et al., 2014). While the majority of YFP insertion lines are homozygously viable (65.5%, 131/200), those that are not homozygously viable were kept over balancer chromosomes. The CPTI identifiers of 200 YFP insertion lines are given in Table S2.

### Whole-mount smFISH and immunofluorescence
*Drosophila* larval CNS and NMJ specimens were prepared using a protocol that was previously described (Titlow et al., 2018). Briefly, specimens were fixed in PFA (4% in PBS with 0.3% Triton X-100 [PBTX]) for 25 min, rinsed three times in PBTX, blocked for 30 min in PBTX + BSA (1%), and incubated overnight at 37°C in hybridization solution (2× SSC, 10% formamide, 10% dextran-sulfate, smFISH probes [250 nm; individual probe sequences are listed in Table S5], and primary antibodies). The next morning, samples were rinsed three times in smFISH wash buffer (2× SSC + 10% formamide) and incubated for 45 min at 37°C in smFISH wash buffer with secondary antibodies and DAPI (1 µg/ml) and then washed for 30 min in smFISH wash buffer at room temperature before mounting in glycerol (Vectashield). PBTX was used in place of smFISH wash buffer for experiments that did not require smFISH. The following antibodies were used: mouse anti-Dlg1 (1:500; 4F3, Developmental Studies Hybridoma Bank), HRP-Dylight-405/488/Alexa Fluor 568/Alexa Fluor 659 (1:100; Jackson ImmunoResearch Laboratories), donkey anti-guinea pig Alexa Fluor 488 (1:500; Thermo Fisher Scientific), and donkey anti-mouse Alexa Fluor 568 (1:500;

Thermo Fisher Scientific). We estimate that the efficiency of detection of individual mRNA molecules was 85% in the brain and 78% in the NMJ (Fig. 1 E''). Although we anticipate that the exact efficiency of detection varies considerably between experiments, these efficiencies are likely to be conservative estimates, and in many individual experiments, the sensitivity is likely to be higher. Measuring the sensitivity for every individual experiment is not practical.

### Image acquisition, postprocessing, and analysis
Whole-mount immunofluorescence and smFISH specimens were imaged on a spinning-disk confocal microscope (Ultraview VoX; PerkinElmer) with 60× oil objective (1.35 NA, UPlan SApo; Olympus) and electron-multiplying charge-coupled device camera (ImagEM; Hamamatsu Photonics) or laser scanning confocal microscope (Olympus Fluoview 3000, 1.30 NA SI UPLASA-PO60XS2, GaSP detector; or Zeiss LSM880, 63× 1.4 NA oil objective, GaSP detector). Images were acquired using Volocity (Perkin Elmer) or FV31S-SW (Olympus) software. Specimens were mounted in Vectashield (H-1000) for imaging following fluorochromes: DAPI, DyLight 405 (HRP), Venus YFP, Quasar 570 (*syp* smFISH probes), and Quasar 670 (*YFP* smFISH probes). Consistent image acquisition settings (laser power, detector gain, pixel dwell time, camera exposure, or temperature at 20–21°C) were used for experimental and control experiments. Acquisition settings were optimized to achieve fast imaging and high signal:background for each instrument. For imaging at the NMJ, the spatial resolution of the microscope combined with the clear delineation of axon and PSD compartments by the two markers provide sufficient discrimination to support the qualitative conclusion that molecules are present in either or both compartments.

Minimal postprocessing was performed on the images. Raw data were analyzed for manual scoring. For quantitative analysis, background subtraction was performed with the rolling ball subtraction algorithm in ImageJ (radius = 5 pixels for smFISH data and radius = 20 pixels for cell markers).

### Spaced potassium stimulation protocol
Third-instar *Drosophila* larvae were dissected in two separate chambers (35 mm Sylgard elastomer-lined Petri dishes) to allow even saline perfusion from peristaltic pumps. A series of five short high potassium saline (KCl, 90 mM) pulses (2, 2, 2, 4, and 6 min) were separated by 15-min perfusion of HL3 saline as described previously (Ataman et al., 2008). For smFISH and immunofluorescence, the larvae were fixed 150 min after the first stimulus and images were acquired on the spinning-disk confocal system described in the section above. For live imaging experiments, images were acquired on the Zeiss LSM880 system described above (20× 1.0 dipping objective) from 10 min after the last stimulus.

### Software pipeline for browsing and annotating the dataset
We built a generalizable pipeline to display high-resolution microscopy images, annotate specific features, and browse the collection from each gene together with relevant publicly available data. An overview of the pipeline is shown in Fig. 2. Raw image data were uploaded to an OMERO server where

multichannel figures displaying multiple fields of view were generated for each image and displayed in OMERO.Figure at the appropriate image plane. After creating a separate figure for each gene in various nervous system compartments, figures were extracted as .jpg files that were used both for annotation and to build a browsable image analysis platform in MDV. To enrich the image collection, we added phenotypic and physical information corresponding to each gene and expanded this information to include the whole genome in an effort to impute our screening results to genes with similar characteristics. Included in the pipeline is a Python script that extracts user-specified gene data from Intermine and local.csv files.

### Annotation comparison and conflict resolvement

Multiple members of the Davis lab with expert knowledge of the larval nervous system tissue and smFISH signal interpretation annotated the figures. Each figure was annotated by three different scorers and the annotations were compared using a Python script that we wrote in-house. A majority vote was used to resolve any conflicts between the answers selected by the experts. When a majority view could not be reached, a fourth expert was required to resolve the specific conflict by focusing with more time on the particular set of images alone. This approach ensured the high confidence and quality of all our annotations.

### Annotate.OMERO.Fig

To facilitate image annotation, we built a Python application with a graphical user interface using the PyQt5 library. The app makes it easy to systematically cycle through a large set of images and score them based on a list of user-defined questions with true/false, multiple choice, or write-in answers. The output is a .csv file that can be directly analyzed or uploaded together with phenotype and gene information to an MDV database. The GitHub repository submitted with this manuscript contains code that interfaces with the Intermine Python API to append Flymine queries and to local file directories to append various public datasets from different model organisms.

### Statistical analysis of ghost bouton, mRNA, and protein levels

Statistical tests that were applied to each dataset are given in the figure legends along with the number of samples appearing in each graph. The normality assumption was tested with the Shapiro–Wilk test. The equal variances assumption was tested with an F test or Levene's test depending on the number of groups. Normally distributed populations with equal variances were compared using Student's $t$ test or one-way ANOVA (with Tukey test for multiple comparisons) depending on the number of groups. Populations with nonnormal distributions were compared using the Wilcoxon rank sum test or Kruskal–Wallis test (with Dunn's test for multiple comparisons) depending on the number of groups. All statistical analyses were performed in R (v3.3.2 running in Jupyter Notebook).

### Lead contact

Further information and requests for resources and reagents should be directed to and will be fulfilled by the lead contact, Ilan Davis (ilan.davis@bioch.ox.ac.uk).

### Online supplemental material

Fig. S1 shows that YFP insertion does not affect the localization and expression of the tagged Dlg1 protein and RNA. Fig. S2 illustrates that expression and physical properties of screened genes are mostly representative of the whole transcriptome. Adding to Fig. 4, Fig. S3 shows *indy*, a highly transcribed gene in the developmental stage of the neuroblast lineage. Fig. S4 presents a GO enrichment analysis of genes with discordance between mRNA and protein expression in specific tissues. Related to Fig. 6, Fig. S5 shows that for the *Gs2* gene, the mRNA is localized in glial boutons at the larval NMJ. Likewise, Fig. S7 shows enrichment of protein localization of the gene *RnpS1* at the NMJ PSD. Fig. S6, related to Fig. 8, examines the effect of Kinesin Heavy Chain absence of transport on *sgg* mRNAs. Table S1 lists GOSlim terms that correspond to the 200 genes examined. Table S2 shows the summary of the annotations for the surveyed 200 genes by the team of experts. Table S3 shows protein and mRNA localization in glia, neuroblast lineage, and PSD. Table S4 provides the data used in the MDV database. Table S5 lists the oligonucleotide sequences used for smFISH in this study.

### Data availability

All Python code is freely available at the GitHub repository: https://github.com/ilandavislab/Annotate.OMERO.Fig.
An MDV instance with our dataset as well as guidance on how to use MDV and browse the collection together with associated information can be accessed at https://doi.org/10.5281/zenodo.6374011.

## Acknowledgments

We are very grateful to the Bloomington, Vienna, and Kyoto Drosophila Stock Centres (fly stocks), Flybase and Flymine (Lyne et al., 2007) for their reagents and open data, which were invaluable to this work. We are grateful to David Ish-Horowicz, Alfredo Castello, and members of the Davis laboratory for critical reading of the manuscript and feedback on the project. We thank Zegami Ltd. for their help, advice, and hosting the collection.

This work was generously supported by a Wellcome Senior Research Fellowship (096144) and Wellcome Investigator Award (209412) to I. Davis, which funded A.I. Järvelin, R.M. Parton, J.S. Titlow, and M.K. Thompson. Advanced microscopy facilities and technical advice as well as support to D.M. Susano Pinto were provided by Micron Oxford (https://micronoxford.com), supported by Wellcome Strategic Awards (091911 and 107457) and a Medical Research Council/Engineering and Physical Sciences Research Council/Biotechnology and Biological Sciences Research Council next-generation imaging award to I. Davis as the principal investigator. J.S. Titlolw and M.K. Thompson were supported by a Leverhulme Trust grant to I. Davis. Department of Biochemistry DPhil studentships supported J.Y. Lee and D.S. Gala. M. Kiourlappou was supported by the Biotechnology and Biosciences Research Council, grant numbers: BB/M011224/1 and BB/S507623/1, by A.G. Leventis Foundation, and by Zegami Ltd.

Author contributions: J.S. Titlow: Conceptualization, Methodology, Software, Formal analysis, Investigation, Data Curation, Writing - Original Draft, Writing - Review & Editing, Visualization, Supervision, and Project administration. M. Kiourlappou: Methodology, Software, Formal analysis, Data Curation, Writing - Review & Editing, Visualization, and Project administration. A. Palanca: Methodology, Investigation, Data Curation, Writing - Review & Editing, Visualization, and Project administration. J.Y. Lee: Formal analysis, Investigation, Data Curation, Writing - Review & Editing, Visualization, and Project administration. D.S. Gala: Validation, Investigation. D. Ennis: Formal analysis, Investigation, Data Curation, Writing - Original Draft, Writing - Review & Editing, Supervision, and Project administration. J.J.S. Yu: Investigation. F.L. Young: Investigation. D.M. Susano Pinto: Software. S. Garforth: Investigation. H.S. Francis: Investigation. F. Strivens: Investigation. H. Mulvey: Investigation. A. Dallman-Porter: Investigation. S. Thornton: Investigation. D. Arman: Investigation. M.J. Millard: Investigation. A.I. Järvelin: Formal analysis, Visualization. M.K. Thompson: Software. M. Sargent: Software. I. Kounatidis: Investigation. R.M. Pargent: Resources. S. Taylor: Software. I. Davis: Conceptualization, Methodology, Formal analysis, Resources, Writing - Original Draft, Writing - Review & Editing, Supervision, Project administration, and Funding acquisition.

Disclosures: I. Davis is on the Scientific Advisory Boards of Open Microscopy Environment and Zegami. S. Taylor is the founder of Zegami. M. Kiourlappou is partly funded by Zegami Ltd., acting as an industrial partner for an iCASE studentship. These organizations did not have a role in the study design or interpretation of its findings. The authors declare no competing financial interests with respect to Zegami or any aspects of the study and manuscript.

Submitted: 28 May 2022

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

# Supplemental material

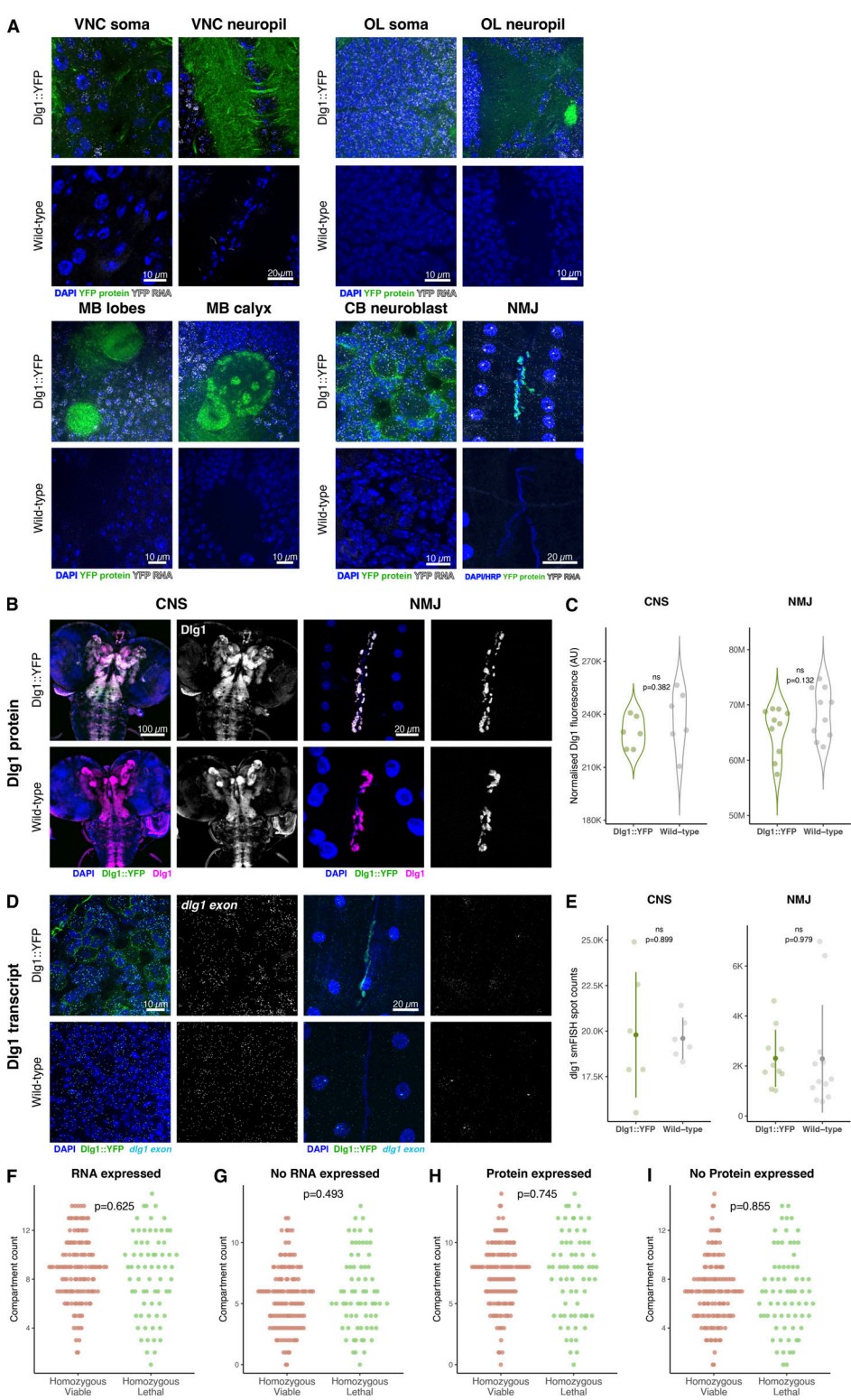

Figure S1. **YFP insertion does not affect the localization and expression of the tagged Dlg1 protein and RNA. (A)** Expression and localization of tagged Dlg::YFP protein and mRNA in CNS and NMJ compartments of Dlg::YFP and wild-type lines. Representative 5 µm maximum projected confocal images are shown. Note that the central brain (CB) neuroblast image with Dlg1::YFP (bottom right set of panels) was taken from the same brain region as in Fig. 1 A, but these are completely separate samples that serve as different controls. Here, we are comparing fluorescent background (negative control), and in Fig. 1 A we show codetection of the same transcript with different probes (positive control). MB, mushroom body; OL, optic lobe. **(B and C)** Comparison of the level of Dlg1 protein in the CNS (100 µm max projected) and NMJ (10 µm max projected) between Dlg::YFP and wild-type flies. **(D and E)** Comparison of the level of dlg1 transcript in the CNS and NMJ (5 µm max projected) between Dlg::YFP and wild-type flies. Data are shown as mean ± SD. **(F–I)** Comparison of the number of nervous system compartments with or without protein and mRNA expression between homozygous viable (131 lines) and lethal (69 lines) CPTI insertions (two-sample Wilcoxon rank-sum test).

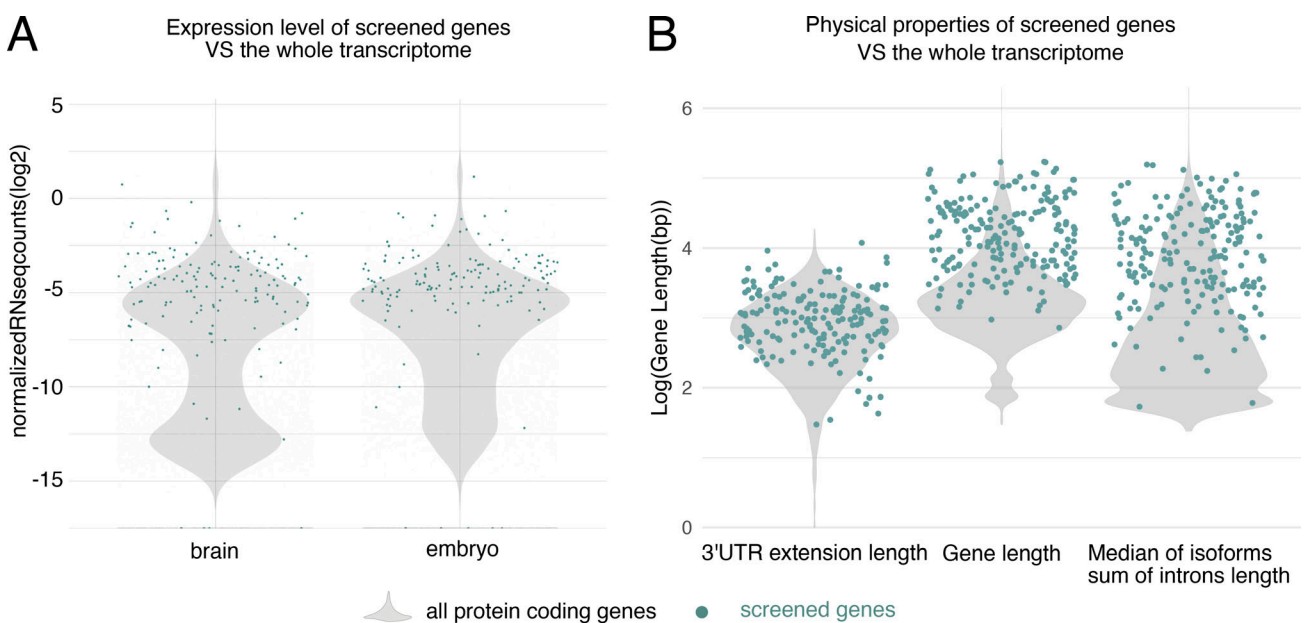

**Figure S2.** **Expression and physical properties of screened genes are mostly representative of the whole transcriptome.** Previously published datasets (Flymine) were used to determine if the collection of screened genes show any biases relative to the rest of the genome. **(A)** Distribution of expression levels in the screened genes are similar to the rest of the transcriptome in both the brain, and in the embryo. **(B)** While the length of 3'UTR extension in the screened genes is similar to the rest of the transcriptome, the screened genes on average have a longer total length and longer introns.

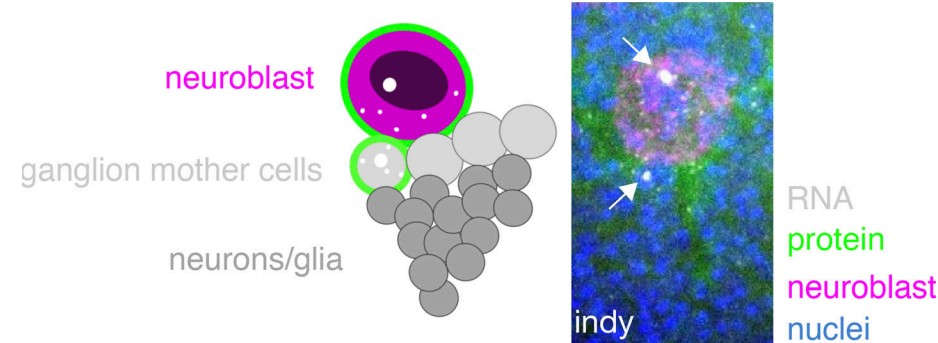

**Figure S3.** **Rare example of highly developmental stage-specific gene expression in the neuroblast lineage.** *Indy*, a plasma membrane transporter for Kreb's cycle intermediates, is highly transcribed (note arrows pointing to large, white punctae) in the neuroblast and a single ganglion mother cell (GMC). *Indy* mRNA and protein are found at low levels in older GMCs, but not in the differentiating neurons/glia.

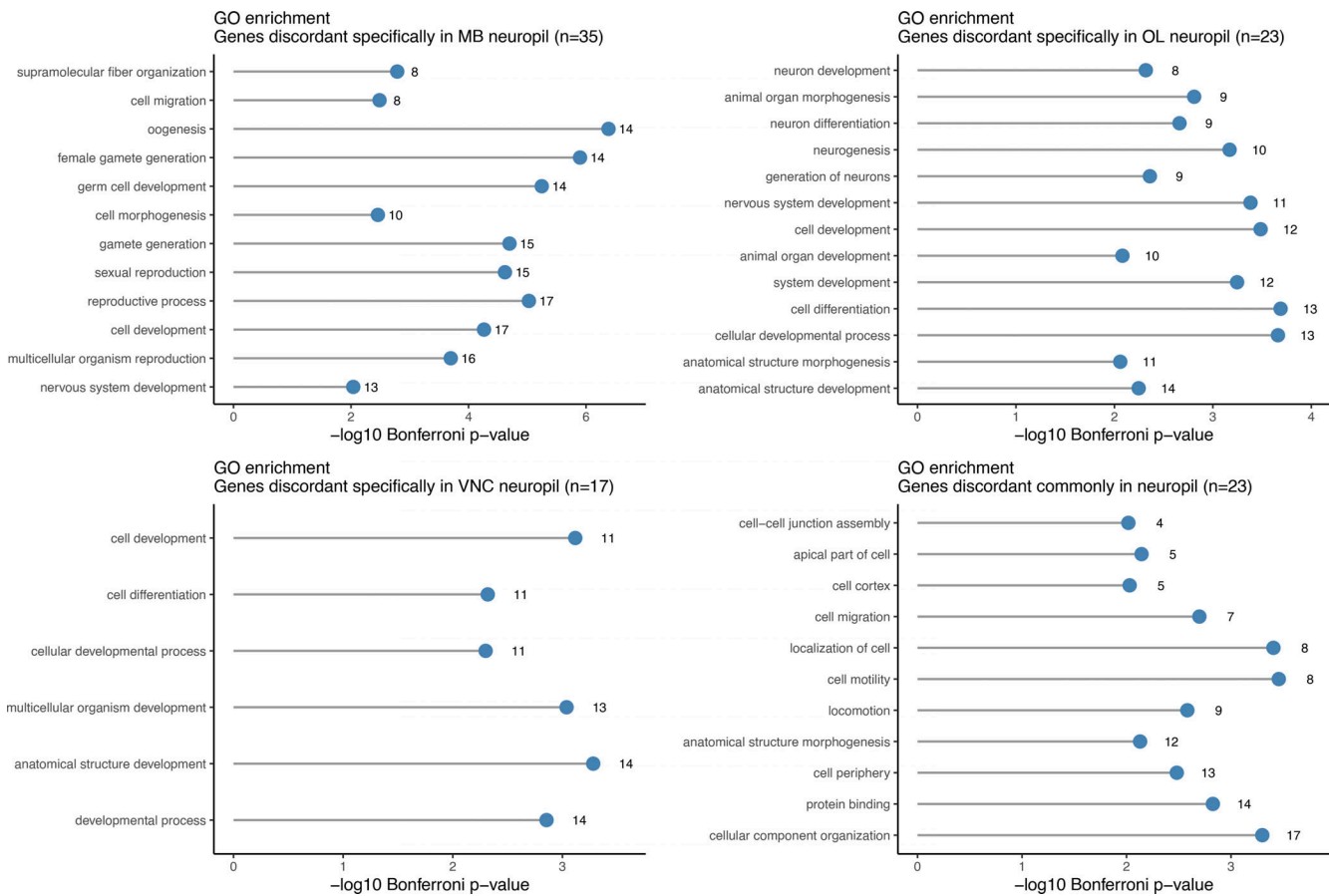

Figure S4.  **GO enrichment analysis of genes with discordant RNA and protein expression in specific synaptic neuropils.** GO enrichment analyses were performed for sets of genes that were discordant in each compartment, and for genes with discordant RNA and protein expression across all synaptic neuropil compartments. Only statistically significant categories are shown.

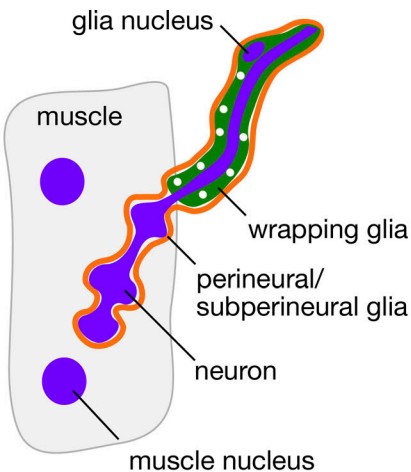

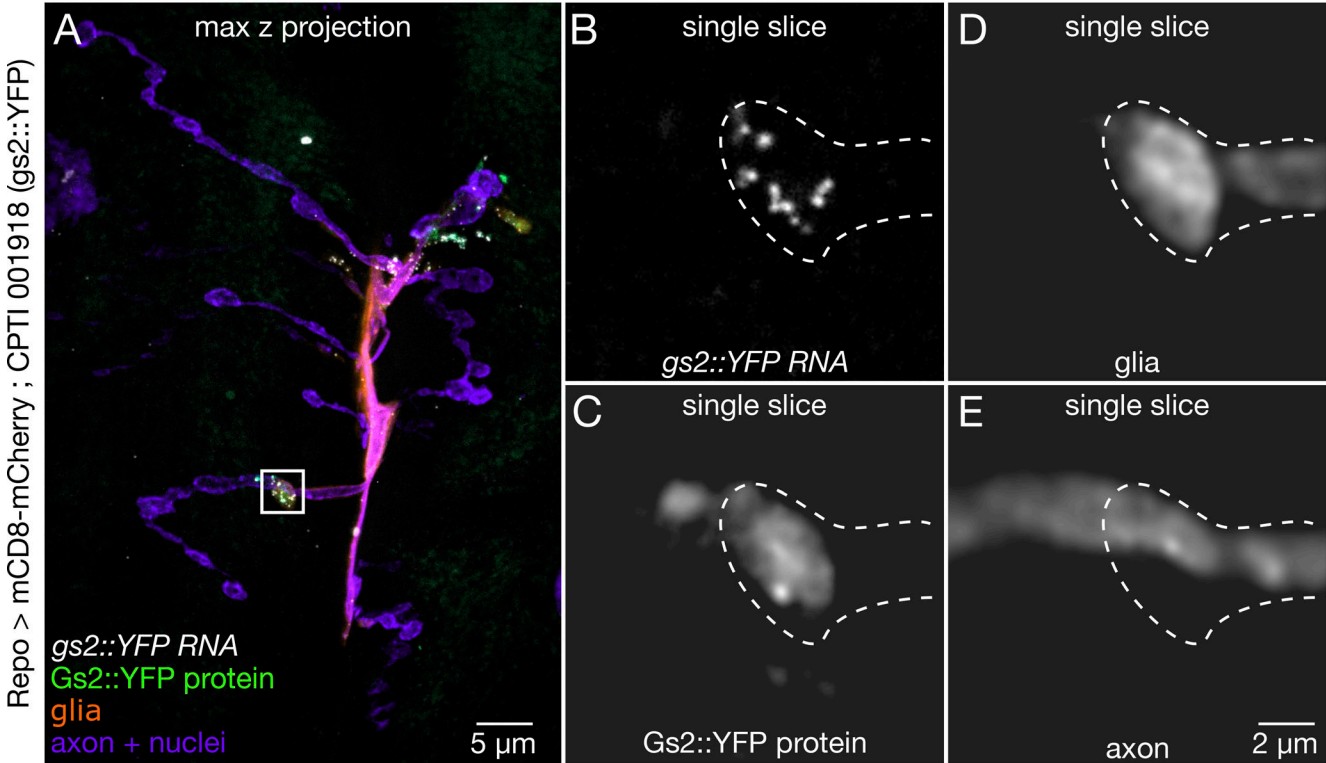

Figure S5. ***Gs2 mRNA is localized in glial boutons at the larval NMJ.*** Representative confocal images showing *gs2::YFP* protein and mRNA expression (white) at the NMJ with markers for glia (*repo > mcd8-mCherry*, orange) and neurons (HRP, purple). **(A)** Full field of view image shows *gs2* expression is confined to proximal regions of the axon terminal where the glial reporter is located. **(B–E)** Magnified regions of a single optical slice from A (white box), showing several molecules of *gs2* mRNA and protein in a perineural glial process enveloping a bouton.

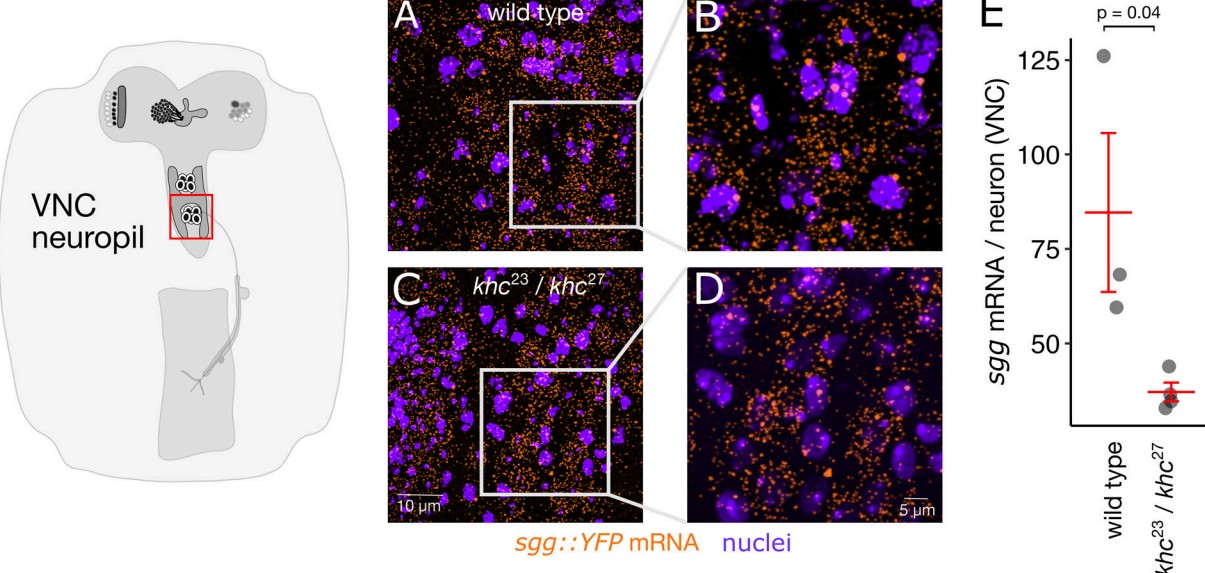

**Figure S6.  Loss of Kinesin Heavy Chain does not cause *sgg* mRNAs to "pile up"' in the motor neuron soma. (A)** Max intensity projection of *sgg* smFISH (orange) and nuclei (purple) signal in a few segments of a wild-type VNC. **(B)** ROI (250 × 250 pixels) that was used for quantifying the number of *sgg* mRNA/neuron. **(C and D)** Max intensity projections of *sgg* mRNA and nuclei signal from a *khc²³/khc²⁷* mutant VNC. **(E)** Quantification shows that there are significantly fewer *sgg* mRNA/cell in *khc²³/khc²⁷* mutant VNCs (unpaired *t* test).

## NMJ - neuromuscular junction

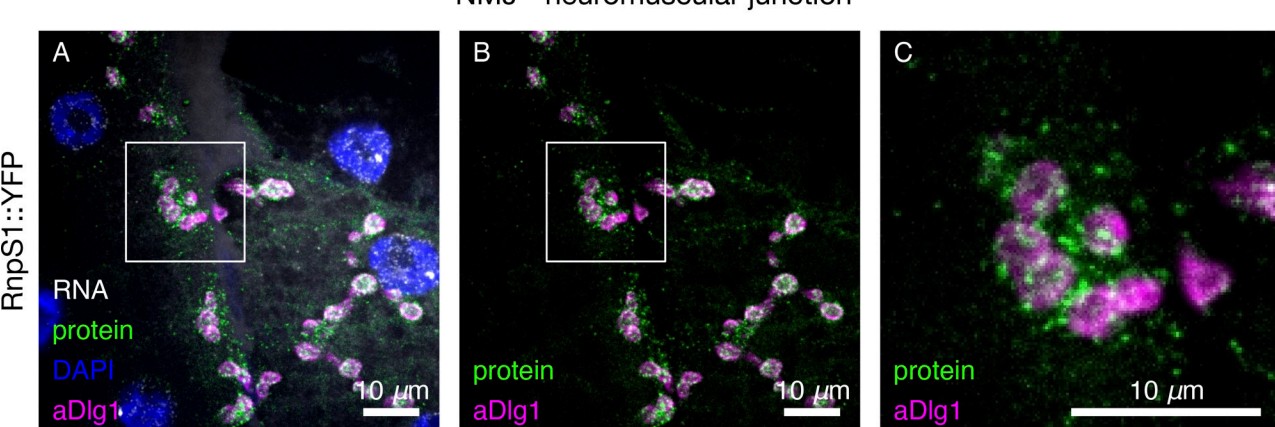

**Figure S7.  Enrichment of protein localization at the NMJ PSD. (A–C)** Maximum intensity projection of confocal image showing *RnpS1::YFP* mRNA and protein localization at the larval NMJ. Enrichment of RnpS1::YFP protein is observed as discrete punctate particle in close proximity to the NMJ PSD, which is labeled using anti-Dlg1.

**Provided online are five tables. Table S1 lists GOSlim coverage of the surveyed 200 genes. Table S2 is a summary of the localization scoring survey. Table S3 shows protein and mRNA localization in glia, NB lineage, and PSD. Table S4 shows accompanying data for MDV database. Table S5 shows smFISH probe sequences used in this study.**

