## [Peer Review File · The Journal of Cell Biology]

Systematic analysis of YFP gene traps reveals common mRNA/protein discordance in neural tissues

Joshua Titlow, Maria Kiourlappou, Ana Palanca, Jeffrey Lee, Dalia Gala, Darragh Ennis, Joyce Yu, Florence Young, David Miguel Susano Pinto, Sam Garforth, Helena Francis, Finn Strivens, Hugh Mulvey, Alex Dallman-Porter, Staci Thornton, Diana Arman, Aino Järvelin, Mary Thompson, Ilias Kounatidis, Richard Parton, Stephen Taylor, and Ilan Davis

Corresponding Author(s): Ilan Davis, University of Oxford

Review Timeline:

Submission Date:	2022-05-28
Editorial Decision:	2022-06-12
Revision Received:	2022-11-11
Editorial Decision:	2022-11-15
Revision Received:	2022-11-28

Monitoring Editor: Juergen Knoblich

Scientific Editor: Dan Simon

Transaction Report:

DOI: <https://doi.org/10.1083/jcb.202205129>

Revision 0

Review #1

1. Evidence, reproducibility and clarity:

Evidence, reproducibility and clarity (Required)

This manuscript by Titlow et al. systematically analyzed spatial distribution of 200 gene's mRNA and protein, and found common discordance between them. Moreover, the browsable resource is pretty useful to most fly people. Though the authors did huge amount of experiments and analysis, and got several really interesting findings, there are some basic questions need to be answered.

Major 1: For the wildtype CS flies, there is no YFP mRNA signal in neuroblast region and how about YFP mRNA signal in MB, OL VNC and NMJ regions? What is the criterion of setting laser power and gain for the mRNA level of 200 genes? Is it difficult to distinguish background and true signal of the mRNA in different area?

Major 2: Would the insertion of YFP affect gene expression? Comparing to CS in Fig 1K, the *dlg1* mRNA signals in *dlg1::YFP* line (Fig 1F) increases a lot. I do not know if this phenotype happens only in this area. So could you show some other regions for *dlg1::YFP* flies.

Major 3: Is the *dlg::YFP* homozygous available? Among 200 gene trap lines, how many of them can be homozygous?

Major 4: Have you tried to investigate the mRNA and protein localization in adult brains?

Major 5: In Fig 3C, the authors claimed in MB or OL soma regions, some genes are protein expression only but no mRNA present. I wonder how do you explain this phenotype in soma.

Major 6: Since *sgg* mRNA localize to both sides of NMJ, would KCl stimulus affect *sgg* mRNA amount and localization in muscle?

Minor 1: You claimed that Fig 1E shows high magnification image of the inset in D, but the scale bars are the same.

Minor 2: Figure 1 legend: K-N, are the images individual channels shown in E? Or in J?

Minor 3: In Fig 2A, optic lobe neuropil and VNC neuropil are mislabeled.

Minor 4: Only one panel has scale bar in Fig 4.

Minor 5: What is Fig 5B'and F'? You should describe them in the Figure legends.

2. Significance:

Significance (Required)

The browsable resource is pretty useful to most fly people. The authors did huge amount of experiments and analysis, and got several really interesting and important findings. This work will provide mRNA localization information for post-transcriptional regulation studies.

3. How much time do you estimate the authors will need to complete the suggested revisions:

Estimated time to Complete Revisions (Required)

(Decision Recommendation)

Less than 1 month

4. Review Commons values the work of reviewers and encourages them to get credit for their work. Select 'Yes' below to register your reviewing activity at Publons; note that the content of your review will not be visible on Publons.

Reviewer Publons

Yes

Review #2

1. Evidence, reproducibility and clarity:

Evidence, reproducibility and clarity (Required)

****Summary****

Titlow et al present a data resource paper for mRNA localization and protein expression in vivo focusing on the larval nervous system which is an area of high interest currently. They screen a known group of YFP gene trap lines (200 lines) and looked at specific aspects of the nervous system such as expression in neuroblasts, the mushroom bodies, glia or the NMJ. They also

present a computational workflow using this set of 200 genes for the investigation of the subcellular localization and potential role of post transcriptional regulation in whole larval tissues. This uses the image data obtained experimentally and then compares with existing datasets to obtain more information.

****Major comments****

The authors results largely support the claims made in the manuscript. Is a clear proof of concept analysis of specific examples and then presentation of examples from different part of the nervous system. Different aspects of the gene trap lines are taken into account. Is a high level analysis of the sub cellular localization of mRNA and protein in different parts of the nervous system. Some interesting new insights which can lead to more in depth analysis of mechanism are presented. Is an interesting idea and presents a method in which to approach a field that has many remaining open questions. This manuscript is an important and timely analysis that will be of high interest in the field.

Is a positive that the authors confirmed the YFP mRNA in situs with an endogenous gene in situ. Although the group is using an established and published set of gene traps, it would be good to confirm protein expression for same gene to increase confidence or provide more details on how is known that the YFP insertions do not affect mRNA stabilization or transcription or protein expression/localization. For example in Figure 1 F' versus K it is unclear why in the DlgYFP insertion there are more Dlg in situ signals than are observed in and around a neuroblast as compared to the wild type control. From the description provided these appear to the maximum intensity images. Is this due to background or an effect of the YFP insertion itself? Because of the increased level of expression is there a feedback loop of the protein regulating the mRNA expression? If had expression of Dlg protein in this figure would also confirm the YFP insertion mirrored the endogenous and it would be easier to discern if there were any changes in the number of Dlg mRNA molecules present. As this was the proof of principle example for the screen this information would increase confidence in the remainder of the data presented. AS an important part of the screen is looking at the potential for post transcriptional regulation this is an important factor to address

Will this pipeline capture information on whether is secreted (contain a signal regulatory peptide) or not as then would expect to be discordant. This should be clarified or commented on. General molecular function is listed in supplementary table 1 but will other types of information be able to be correlated from datasets or databases as well.

****Minor comments****

On page 9 refer to Figure 6S which I think is supposed to be Figure S6. In text refer to an example of gli but show gs2 in the figure so it is unclear what is being referred to or shown. Could include more description on the generation of the supplementary tables and analysis of the tables. I could not find any description/legend which made analysis of some of the tables more difficult.

The data set was trained on a known set of data (analyzed by experts. It would be interesting to see what it could do with a novel set of genes in the context of post transcriptional regulation, but that is beyond the overall scope of this manuscript.

2. Significance:

Significance (Required)

This is an interesting idea and is a useful resource for the genes analyzed. Gives an initial tool to analyze the expression of genes. Allows for systematic analysis of mRNA (smFISH) and protein on a larger scale but with high resolution. Adds new knowledge in terms of the localization of mRNAs and protein in the periphery of neural and glia processes which may inform future analyses of the role of these genes in these tissues.

Is a useful resource within neurodevelopment in Drosophila and post transcriptional regulation. Would be of interest to a general audience as workflow could be applied to any tissue or set of genes. Covers a very broad set of genes with disparate biological functions again making this of interest to a broader audience.

Expertise of reviewer

Drosophila, neurodevelopment, RNA regulation, post transcriptional regulation, polarity and adhesion.

3. How much time do you estimate the authors will need to complete the suggested revisions:

Estimated time to Complete Revisions (Required)

(Decision Recommendation)

Between 1 and 3 months

4. Review Commons values the work of reviewers and encourages them to get credit for their work. Select 'Yes' below to register your reviewing activity at Publons; note that the content of your review will not be visible on Publons.

Reviewer Publons

No

Review #3

1. Evidence, reproducibility and clarity:

Evidence, reproducibility and clarity (Required)

In this manuscript, the authors address the important topic of post-transcriptional gene regulation using the larval nervous system in *Drosophila*. They utilize a novel approach taking advantage of existing protein trap library, which permits use of the same smFISH probe to detect an array of 200 RNAs and visualize their corresponding protein expression. Furthermore, the authors developed a computational pipeline to visualize and analyze the resulting data, which should enhance the application of this method by other researchers. A major strength of the data comes from the analysis of multiple cell types in distinct compartments of the nervous system, cell types (neuron, glia, neuroblast), and subcellular domains. From the cumulative data, the authors are able to describe several interesting observations relating to cell-specific post-transcriptional regulation, regulation within a central-neuroblast lineage and glial post-transcriptional regulation, among others.

However, in spite of these strengths, there are several concerns related to the organization and interpretation of the manuscript that the authors should address in order to improve the manuscript:

****General concerns:****

1. The approach relies on gene traps that often fail to be made homozygous, presumably due to deleterious function of the YFP insert. This is an obvious limitation of the study, which the authors address, but do so insufficiently by only analyzing a single case *Dlg1*. The authors should report how many of the 200 YFP-traps can produce viable homozygous animals, whether phenotypes can be observed, and any other relevant information to assess the functional properties of the tagged genes.
2. The term "discordant" is used for non-congruous RNA/Protein levels in soma and distal processes, and sometimes the two are analyzed in the same figure (e.g Fig 3A). When it is stated that 98% of genes are discordant, this is an over-simplification as what the authors describe as "discordant" is expected to occur frequently in the distal process, but less often in the soma (which is what the authors find when presenting the data for individual compartments - Fig 3B-C). This is confusing because the observation means completely different things in the two compartments, though both are interesting to describe. These analyses, and their interpretation, should be kept separate.
3. There is not enough emphasis placed on the cell-type specific regulation of RNAs. There are very few studies that have investigated how localization of individual RNAs changes in different cell types or regions of the nervous system, and the authors find that this is quite prevalent. Therefore, the rather superficial analysis of these data fails to take advantage of a major strength of the data. For example, for the discordant genes that differ in neuropil localization between different regions of the CNS, what types of molecules do they encode, what is their function in neurons (if known), and why might they be required locally in one region of the CNS but not the other?
4. The authors conclude that mRNA and protein co-localization in glia processes shows that mRNA localization makes a major contribution of the proteome in processes. However, there is

not enough evidence for such conclusion since neither translation of these mRNAs nor lack of protein trafficking from the somas was shown.

5. An important caveat of this technique that should be discussed is the lack of knowledge about the translation of these mRNAs, if the mRNA that is being detected is the same as the one that is translated. While the authors emphasize the discordance between mRNA and protein localization, it is not possible to know whether these mRNAs are being translated where they are found, e.g. soma vs neuropil. Moreover, there are many examples (e.g. BDNF) where the isoform influences the subcellular localization of the mRNA. There is no way of studying the isoforms here, and we could be looking for a different mRNA isoform localized to a specific compartment compared to the protein. These points must be discussed.

****Minor suggestions:****

- The authors should identify GO terms to understand what types of molecules are subjected to RNA regulation. They provide a supplementary table for all genes, but it would be useful to have a chart showing the proportion of different GO terms represented in the overall gene set, genes that show cell-specific regulation, genes that show neuron vs glia specific regulation, etc.

- "However, post-transcriptional regulation can also manifest itself within a cell, so that a protein is localised to a distinct site from the mRNA that encodes it".

While subcellular RNA localization may represent a regulatory layer, I do not agree that proteins that function in the cell at a different location than their translation site represents regulation per se. Many such cases exist for proteins that are trafficked!

- "The majority of individual puncta appearing in the *dlg1::YFP* line (51% in the brain, 64% in larval muscles".

Why is the agreement between YFP and endogenous FISH so low? Do many individual RNAs fail to hybridize? This should be discussed.

- "However, one gene, *indy*, is highly transcribed in neuroblasts and a single ganglion mother cell before it is rapidly shut off (Figure S1A)". This figure does not exist. Where are the data?

- The authors should be consistent about calling perineurial or perineural glia (both correct) in their images and text.

- "We only observe a minority of localised axonal mRNAs that lack the protein they encode at the axon extremities, in contrast to our findings in the mushroom body, optic lobe, and ventral nerve cord neuropils"

These results are not contrasted, as in all neuropils the minority of localized mRNAs are those lacking their corresponding proteins. For example, 9% in NMJ vs 7.5% in OL neuropil according to Fig. 1B. What is conflicting with the conclusion?

- "These results suggest that motor axons are more selective than the other neuronal extensions in the mRNAs that are transported over their very long distances from the soma to the neuromuscular synapse"

The current literature says that the same mechanism (cis-elements) is used to transport mRNAs to subcellular compartments, which would be inconsistent with the idea of motor axons being "more selective" than other neurons for the same mRNA, but just a result of fewer mRNAs being found in motor neurons: 34.% of the mRNAs are found in motor neurons soma vs 83% in OL soma, 86.5% in VNC soma, and 70.5% in MB soma. To get to this conclusion, the authors should show that mRNAs previously found in the neuronal extensions of other neurons are not found in the axons of motor neurons but are still expressed in their somas. They might want to

suggest different RBPs involved in the transport or discussing the very long distance they need to travel which can influence their detection in the tips.

Figures

- Figure 1. Experimental approach summary

- Some colors do not show well and should be changed, e.g: grey in Fig. 1A, and Fig. 1B probe sites indicated in light blue and pink within the introns of *dlg1*.

- Fig. 1E': There appears to be a large discrepancy in co-detection % for CNS and muscle in the graph judging by the size of circles, yet in the text, it is stated that there is average of 51% and 64% in the two, respectively. I don't see any green circles with over 25% agreement in the graph. Are the colors correct here?

- Fig. 1D-I: It's difficult to identify where the zoomed panels come from. E has its own square (indicating zoom in E'). Please make this square dashed or a different color in E so it is clear F and G do not come from there.

- Comparing Fig. 1F vs K: Why does there appear to be so much more *dlg1* mRNA in the YFP-tag condition? If this is due to selection of imaging area, please choose a more similar region to image so the RNA levels are comparable. Otherwise it indicates the YFP-tag line has more RNA expression, which is likely not the case.

- Figure 2. Analysis pipeline overview

- The lines for the first two zoomed panels are switched: The optic lobe is going to VNC and vice-versa.

- Figure 3. Overall summary of results

- Figure 3A: Soma/Neuropil/muscle should be separate or at least ordered such that they are next to each other to facilitate direct comparison of genes in the same region of the cell in neurons from different CNS areas. Why are glia not included in this summary? A third color should be used to indicate when there is neither mRNA nor protein expression.

- "Compiling all the information together shows that there are that 196/200 or 98% of the genes show discordance between RNA and protein expression"

However, 5 genes shown in Fig. 3A do not show "discordance": *CG9650*, *cup*, *Lasb*, *rg*, and *vsg*!!

- Figure 4. Neuroblast lineage analysis

- Is clustering around the NB sufficient to determine lineage relationship? There seems to be other neurons around the NB.

- More examples should be shown for the post-transcriptional category, as it is the most interesting category, and there are many different possible outcomes. Are there cases of transcriptional control and post-transcriptional regulation? Are there cases where the youngest neurons (closer to the NB) in the progeny are expressing the protein while the oldest are not? If not, could this be an artifact from a slow translation and the protein being detected only after building up in the cell? Top1 protein (Fig. 4D) seems to be less expressed in the youngest neurons.

- "The transcription rate of these genes, as indicated by the relative intensity of smFISH nuclear transcription foci, is similar across the neuroblast lineage, however protein signal is only detectable in a minority of the progeny cells (Figure 4E)".

Many nuclei lack clear large spots, but have small spots indicative of RNA; how is this interpreted? Do they lack transcription, or is this due failure of the smFISH to capture all transcription sites?

Were transcripts actually counted to assess cell-specific differences? This should be possible

with smFISH

- Figure 5. RNA synaptic localization

- A have global analysis comparison of all neuropil areas would be welcome in this figure.

- "Surprisingly, another 59 transcripts are present at synapses without detectable levels of protein (Figure 5E-H)"

This text does not correspond to Fig 5E-H but 5I-L. Where is the text about 5E-H?

- For Fig. 5J and 5N RNA appears scattered regularly throughout the entire panel area. How sure are the authors that this is not due to poor signal/noise? For example, perhaps too much probe being used for these targets.

- Fig. 5R is not cited in the text.

- Figure 6. RNA localization in glia

- For Fig. 6B-G it is hard to tell if there is any overlap of the RNA and Glia. Maybe show multiple zoomed-in merged images and/or highlight the structures with lines that are present in all panels.

- For Fig. 6L-O: How reproducible is this small amount of RNA puncta in the NMJ glia? Is this possibly biologically important?

- Why do cartoons labelling subnuclear/perinuclear glia in Fig.6 and Fig.S6 show different localization?

- The cartoons seem to extrapolate from the data: While in Fig 6B-D, we see neither the big bright spot of transcription in the glial nucleus nor as many transcripts in the neuropil, they are both present in the cartoon. In Fig. 6E-G there is no indication of cortical glia soma nor the transcription spot only in glia nuclei.

- "To assess glial localisation for the 200 genes of interest, we used a pan-glial gal4 driving a membrane mCherry marker (repo-GAL4>UAS-mcd8-mCherry) to learn the expression pattern of all glial cells, and then classified the pattern in the YFP lines (without the marker) based on knowledge of that expression pattern. We validated this approach by combining the RFP marker" Did the authors use mCherry or RFP for these experiments? Also, the previous sentence is redundant.

- Figure 7. RNA localization at neuromuscular synapse

- RNA for these genes seems far too spread throughout the muscle to draw any conclusions

- Also with so many RNAs distributed in the muscle, specific localization of RNA molecule to the precise PSD would have no conceivable benefit

- I suggest drawing lines around the protein expression to facilitate visualization of the mRNA localization for panels B, F and J. It is especially hard to conclude anything from panels B and F.

- Light grey with white dots is hard to see in the cartoons

- Figure 8. Role of khc and activity in sgg localization

- Presumably there is a huge number of developmental problems associated with this mutant that could cause decrease in sgg localization

- If the authors include this, then they should characterize the mutant NMJs: what is the change in size, synapse number, etc..

- Is there more sgg accumulated in soma as a result of less transport? Is sgg being expressed at the same level?

- Fig. 8F-H: Why is Dlg1 accumulated in the entire axon, not just the presume synapse?

- Fig. 8J: Why is sgg signal occurring in circles disconnected from the main axon? The authors should show a different image

2. Significance:

Significance (Required)

This is a significant and complex paper that contributes with novel tools to an important issue

3. How much time do you estimate the authors will need to complete the suggested revisions:

Estimated time to Complete Revisions (Required)

(Decision Recommendation)

Between 1 and 3 months

4. Review Commons values the work of reviewers and encourages them to get credit for their work. Select 'Yes' below to register your reviewing activity at Publons; note that the content of your review will not be visible on Publons.

Reviewer Publons

Yes

Revision Plan

Manuscript number: RC-2022-01392R

Corresponding author(s): Ilan Davis

1. General Statements

We thank the reviewers for their constructive and helpful comments on our manuscript. We are delighted to find their consensus that the manuscript represents a useful resource for the *Drosophila* community in particular, and for the fields of neural development and post-transcriptional gene regulation. The following is our detailed responses and plan for how we will address all the major points raised by the reviewers. We also plan to address all minor points fully and have been through them in great detail one by one, so we are confident this is feasible within a reasonable and expected time frame.

2. Description of the planned revisions

Reviewer #1

Major 1: For the wildtype CS flies, there is no YFP mRNA signal in neuroblast region and how about YFP mRNA signal in MB, OL VNC and NMJ regions? What is the criterion of setting laser power and gain for the mRNA level of 200 genes? Is it difficult to distinguish background and true signal of the mRNA in different area?

This is a good point about background intensity levels (from non-specific binding of the YFP smFISH probe) across different tissue regions. We thank the review for raising it. Signal:background decreases with depth in all of the tissues, with superficial cells displaying similarly high signal:background in the CNS and NMJ, while signal:background in neuropil regions of the CNS are slightly lower. To address this point, we plan to include a supplementary figure to show background fluorescence of the smFISH probe across all regions of the CNS and NMJ.

To address the point about image acquisition settings, we will included the following additional information in the Methods section (Page 17):

“Consistent image acquisition settings (laser power, pixel dwell time or camera exposure, detector gain) were used for experimental and control experiments. Acquisition settings were optimized to achieve fast acquisition and high signal:background for each instrument.”

We will add a further explicit explanation to the manuscript referring to previous publications, that the nature of the smFISH method makes it relatively simple to distinguish background from true signal. True punctae have a relatively uniform size, symmetrical shape, and consistent intensity distribution. Whereas background punctae that are either larger than diffraction-limited punctae or have lower intensity can easily be separated from real signal.

Major 2: Would the insertion of YFP affect gene expression? Comparing to CS in Fig 1K, the *dlg1* mRNA signals in *dlg1::YFP* line (Fig 1F) increases a lot. I do not know if this phenotype happens only in this area. So could you show some other regions for *dlg1::YFP* flies.

This is a good point raised by both Reviewer #1 and Reviewer #2 (Major point 1). We agree that a proper quantification of the effect of YFP-insertion will bolster our conclusion, highlighting the utility of protein-trap collections for systematic analysis of post-transcriptional regulations. To address this, we plan to: (i) provide quantifications of *dlg1* transcript expression in the CNS and NMJ and compare the levels between *dlg1::YFP* and wild-type lines, and (ii) provide new figure visuals reflecting our quantification results.

Major 3: Is the *dlg1::YFP* homozygous available? Among 200 gene trap lines, how many of them can be homozygous?

This is a good point raised by both Reviewer #1 and Reviewer #3 (Major point 1). The *dlg1::YFP* (CPTI-000207) line used for the control experiments is homozygous. However, it is a great point that not all of the YFP insertions are homozygous viable. Out of the 200 lines we screened, 131/200 (65.5%) insertions are homozygous viable, whereas 69/200 (34.5%) are homozygous lethal or are unknown. We have addressed this caveat in the Methods section (Page 16) with the following statement:

“The majority of YFP insertion lines are homozygous (65.5%, 131/200), those that are not homozygous viable were kept over balancer chromosomes.”

Our provisional analysis shows that the number of nervous system compartments expressing YFP-fused protein or mRNA are not affected by homozygous lethality. We plan to include this analysis in the revised manuscript.

Major 4: Have you tried to investigate the mRNA and protein localization in adult brains?

Yes, in a related study, we demonstrated that this approach also works in the adult brain (Mitchel et al., 2021, [DOI:10.7554/eLife.62770](https://doi.org/10.7554/eLife.62770)). A systematic analysis of protein and mRNA expression patterns in the adult brain would be highly interesting and is certainly possible, however it is beyond the scope of the manuscript. To address this point, we will cite our related work and emphasise more clearly the wider applicability of our technique.

Major 5: In Fig 3C, the authors claimed in MB or OL soma regions, some genes are protein expression only but no mRNA present. I wonder how do you explain this phenotype in soma.

Our favoured explanation is that protein is more stable than mRNA. Therefore, after the mRNA is translated, it could get degraded while the protein is still present in the cell. We will add text in the relevant section to mention potential differential stability of protein/mRNA.

Major 6: Since sgg mRNA localize to both sides of NMJ, would KCl stimulus affect sgg mRNA amount and localization in muscle?

That is an interesting question. The data in Fig. 8I-J show that there is no additional Sgg::YFP protein accumulation at the muscle post synaptic density in response to KCl stimulus. It's been shown elsewhere (Ataman et al., 2008, [DOI:10.1016/j.neuron.2008.01.026](https://doi.org/10.1016/j.neuron.2008.01.026)) that Sgg protein translocates to the muscle nucleus in response to KCl stimulus. Determining whether that mechanism requires translation of new protein would require a complete new study with translational analysis and would distract from the message of the current study.

Reviewer #2

Major 1: Although the group is using an established and published set of gene traps, it would be good to confirm protein expression for same gene to increase confidence or provide more details on how is known that the YFP insertions do not affect mRNA stabilization or transcription or protein expression/localization. For example in Figure 1 F' versus K it is unclear why in the DlgYFP insertion there are more Dlg in situ signals than are observed in and around a neuroblast as compared to the wild type control. From the description provided these appear to the maximum intensity images. Is this due to background or an effect of the YFP insertion itself? Because of the increased level of expression is there a feedback loop of the protein regulating the mRNA expression? If had expression of Dlg protein in this figure would also confirm the YFP insertion mirrored the endogenous and it would be easier to discern if there were any changes in the number of Dlg mRNA molecules present. As this was the proof of principle example for the screen this information would increase confidence in the remainder of the data presented. AS an important part of the screen is looking at the potential for post transcriptional regulation this is an important factor to address.

Thank you for the valuable suggestion. We agree with the reviewer that the comparison of *dlg1* transcript levels would provide a valuable control. This point was raised by both Reviewer #1 and Reviewer #2. Please see **[Reviewer #1 - Major point 2]** for our response.

Major 2: Will this pipeline capture information on whether is secreted (contain a signal regulatory peptide) or not as then would expect to be discordant. This should be clarified or commented on.

The reviewer's comment is correct. Secreted proteins may show discordant distribution of protein and mRNA between cell types even in absence of post-transcriptional regulations. Note that Shaggy (Sgg) is a secreted protein but we observe that most of the protein products are expressed in the

Revision Plan

same cell as the RNA. We propose to follow the reviewer's suggestion and revise the text to discuss the limitation of our pipeline in identifying proteins regulated via secretory modes.

Major 3: General molecular function is listed in supplementary table 1 but will other types of information be able to be correlated from datasets or databases as well.

This question highlights a major feature of our dataset and associated metadata. The analysis in Supplementary Table 1 is used to assess the functional representation of the 200 genes in our screen against the all known genes. We found that ~90% of GOSlim terms are covered by the 200 genes, highlighting the diversity of our list of genes. On the other hand, our Zegami resource (Accompanying data for Zegami) contains a rich collection of metadata (including the full list of GO terms) associated with each gene in the dataset, and extends that information to the entire genome. We anticipate that the Zegami resource will be a valuable platform to query data from our analysis and other databases. To address this, we plan to: (i) revise the legend for the Supplementary Table 1, and (ii) revise the text to clarify what kind of information is available in our Zegami resource.

Reviewer #3

Major 1: The approach relies on gene traps that often fail to be made homozygous, presumably due to deleterious function of the YFP insert. This is an obvious limitation of the study, which the authors address, but do so insufficiently by only analyzing a single case Dlg1. The authors should report how many of the 200 YFP-traps can produce viable homozygous animals, whether phenotypes can be observed, and any other relevant information to assess the functional properties of the tagged genes.

Thank you for requesting further information on homozygous viability of the YFP-trap collection. This point was raised by both Reviewer #1 and Reviewer #3. Please see **[Reviewer #1 - Major point 3]** for our response.

Major 2: The term "discordant" is used for non-congruous RNA/Protein levels in soma and distal processes, and sometimes the two are analyzed in the same figure (e.g Fig 3A). When it is stated that 98% of genes are discordant, this is an over-simplification as what the authors describe as "discordant" is expected to occur frequently in the distal process, but less often in the soma (which is what the authors find when presenting the data for individual compartments - Fig 3B-C). This is confusing because the observation means completely different things in the two compartments, though both are interesting to describe. These analyses, and their interpretation, should be kept separate.

This is a fair point raised by the reviewer. To address this point we plan to: (i) prepare two separate tables summarising our annotation in soma and neurite compartments, and (ii) revise the text accordingly to explain and discuss how the discordant protein and mRNA expression pattern can arise both within different compartments of a cell or between different cell types in a cell lineage

Major 3: There is not enough emphasis placed on the cell-type specific regulation of RNAs. There are very few studies that have investigated how localization of individual RNAs changes in different cell types or regions of the nervous system, and the authors find that this is quite prevalent. Therefore, the rather superficial analysis of these data fails to take advantage of a major strength of the data. For example, for the discordant genes that differ in neuropil localization between different regions of the CNS, what types of molecules do they encode, what is their function in neurons (if known), and why might they be required locally in one region of the CNS but not the other?

We appreciate that the Reviewer recognizes the power of comparing RNA localization patterns across different brain regions (Figure 5R). We reported on a common set of synaptic mRNAs that encode nuclear proteins across the different regions of the nervous system. Per the Reviewer's suggestion, we have begun to look into region-specific patterns of expression. In Figure 5R, two categories with the largest number of genes are 'protein_MB_syn' and 'protein_OL_syn', which contain proteins that are specific to those regions. However, given the small number of 15-16 genes, gene ontology enrichment analysis has limited power to infer information on the entire genome.

We plan to revise the manuscript:

- 1) to include tables with lists of genes specific to MB and OL regions.
- 2) to revise the manuscript to include in the discussion a caveat of the limited power of analysis based on a small number of genes.

Major 4: The authors conclude that mRNA and protein co-localization in glia processes shows that mRNA localization makes a major contribution of the proteome in processes. However, there is not enough evidence for such conclusion since neither translation of these mRNAs nor lack of protein trafficking from the somas was shown.

The significant role of RNA localisation in shaping the local proteome and performing proteostatic regulation has been studied in detail (Zappulo et al., 2017, von Kugelgen and Chekulaeva 2022 Giandomenico et al., 2022). However, the reviewer's comment is correct that we do not show direct evidence of mRNA translation or protein trafficking. Therefore, we propose to: (i) clarify the text by including the citation of these publications, and (ii) qualify our claim that mRNA localization is a major contribution of the proteome in neurite or glial processes.

Zappulo et al., 2017, [DOI: 10.1038/s41467-017-00690-6](https://doi.org/10.1038/s41467-017-00690-6)
von Kugelgen and Chekulaeva 2022 [DOI: 10.1002/wrna.1590](https://doi.org/10.1002/wrna.1590)
Giandomenico et al., 2022, [DOI: 10.1016/j.tins.2021.08.002](https://doi.org/10.1016/j.tins.2021.08.002)

Major 5: An important caveat of this technique that should be discussed is the lack of knowledge about the translation of these mRNAs, if the mRNA that is being detected is the same as the one that is translated. While the authors emphasize the discordance between mRNA and protein localization, it is not possible to know whether these mRNAs are being translated where they are found, e.g. soma vs neuropil. Moreover, there are many examples (e.g. BDNF) where the isoform influences the subcellular localization of the mRNA. There is no way of studying the isoforms here, and we could be looking for a different mRNA isoform localized to a specific compartment compared to the protein. These points must be discussed.

We agree with the reviewer that our method does not provide information on whether the detected mRNA is being translated in time and space. Elucidating the relative contribution of localised mRNA in shaping the local proteome is not a trivial task and it is being actively investigated in the field. However, we believe our dataset provides a unique high-resolution map of transcripts that are potentially regulated at post-transcriptional and translational levels. It would be promising to follow up the 'discordant' genes identified from our survey using experimental methods that are able to track mRNA-ribosome associations (e.g. TRICK) in future studies. To address this point, we will revise the text to discuss this caveat.

Thank you for pointing out the matter with mRNA isoforms. Our preliminary analysis indicates that 71% of the screened genes have constitutive YFP-insertions (i.e. YFP-cassette traps all mRNA isoforms). However, we agree that our approach cannot discriminate the case where protein produced from an mRNA isoform is trafficked and co-localises with another mRNA isoform that did not give rise to that protein. We plan to revise the text to discuss this point explicitly.

3. Description of the revisions that have already been incorporated in the transferred manuscript

Several minor comments regarding typos and simple errors have already been incorporated in the transferred manuscript. The changes are highlighted in yellow in the revised submission.

We plan to address all the useful numerous minor comments that the reviewers have kindly highlighted to us. We feel these are straightforward to do and feasible in a short time, so do not require a detailed listed plan. If the reviewers feel they do afterall need such a list, we will be happy to provide it. However, there is one minor comment that we feel requires a little more explanation:

4. Description of analyses that authors prefer not to carry out

Reviewer #3 - Minor Comment on Figure 8: "...they should characterize the (*khc*) mutant NMJs: what is the change in size, synapse number, etc..

Revision Plan

The *khc* mutants are already known to show synapse morphology phenotypes (Kang et al., 2014), though the *khc²³/khc²⁷* transheterozygous allele has previously been used to assess localization defects at the larval NMJ (Gardioli and St. Johnston, 2014). Moreover, our manuscript (Figure 8) focuses on post-developmental stimulus-dependent processes, rather than cellular-level synapse developmental parameters with this mutant. The reviewer correctly points out that the *khc* developmental phenotypes are likely to have other secondary defects as a result of impaired microtubule transport. The purpose of that mutant was to assess the molecular-level question of whether microtubule-based transport is required for *sgg* mRNA localization at the axon terminal. The consequences and exact mechanism of disrupted transport are beyond the scope of this study. To address this point explicitly, we will:

1. Revise the manuscript to quote more explicitly and clearly the developmental *khc* phenotype.
2. Revise the manuscript to explain the difference between the developmental role of *khc* and role in the transport of *sgg* specifically to the axon terminal.
3. Revise the manuscript to explain more explicitly the limitations of this mutant.

June 12, 2022

Re: JCB manuscript #202205129T

Prof. Ilan Davis
University of Oxford
Department of Biochemistry
South Parks Road
Oxford OX1 3QU
United Kingdom

Dear Prof. Davis,

Thank you for submitting your manuscript titled "Systematic analysis of YFP gene traps reveals common mRNA/protein discordance in neural tissues." We have now assessed your manuscript, the Review Commons reports, and your response to the reviewer comments. We invite you to submit a revision as outlined in your revision plan. We also agree that additional characterization of NMJs is not necessary.

GENERAL GUIDELINES:

Text limits: Character count for a Tool is < 40,000, not including spaces. Count includes title page, abstract, introduction, results, discussion, and acknowledgments. Count does not include materials and methods, figure legends, references, tables, or supplemental legends.

Figures: Tools may have up to 10 main text figures. Figures must be prepared according to the policies outlined in our Instructions to Authors, under Data Presentation, <https://jcb.rupress.org/site/misc/ifora.xhtml>. All figures in accepted manuscripts will be screened prior to publication.

*****IMPORTANT:** It is JCB policy that if requested, original data images must be made available. Failure to provide original images upon request will result in unavoidable delays in publication. Please ensure that you have access to all original microscopy and blot data images before submitting your revision. ***

Supplemental information: There are limits on the allowable amount of supplemental data. Tools generally have up to 5 supplemental figures, you currently exceed this limit but, in this case, we will be able to give you the extra space. Up to 10 supplemental videos or flash animations are allowed. A summary of all supplemental material should appear at the end of the Materials and methods section.

Please note that JCB now requires authors to submit Source Data used to generate figures containing gels and Western blots with all revised manuscripts. This Source Data consists of fully uncropped and unprocessed images for each gel/blot displayed in the main and supplemental figures. If your revised paper will contain cropped gel and/or blot images, please be sure to provide one Source Data file for each figure that contains gels and/or blots along with your revised manuscript files. File names for Source Data figures should be alphanumeric without any spaces or special characters (i.e., SourceDataF#, where F# refers to the associated main figure number or SourceDataFS# for those associated with Supplementary figures). The lanes of the gels/blots should be labeled as they are in the associated figure, the place where cropping was applied should be marked (with a box), and molecular weight/size standards should be labeled wherever possible. Source Data files will be made available to reviewers during evaluation of revised manuscripts and, if your paper is eventually published in JCB, the files will be directly linked to specific figures in the published article.

The typical timeframe for revisions is three to four months. While most universities and institutes have reopened labs and allowed researchers to begin working at nearly pre-pandemic levels, we at JCB realize that the lingering effects of the COVID-19 pandemic may still be impacting some aspects of your work, including the acquisition of equipment and reagents. Therefore, if you anticipate any difficulties in meeting this aforementioned revision time limit, please contact us and we can work with you to find an appropriate time frame for resubmission. Please note that papers are generally considered through only one revision cycle, so any revised manuscript will likely be either accepted or rejected.

Thank you for this interesting contribution to Journal of Cell Biology. You can contact us at the journal office with any questions, cellbio@rockefeller.edu or call (212) 327-8588.

Sincerely,

Juergen Knoblich, PhD
Monitoring Editor
Journal of Cell Biology

Dan Simon, PhD
Scientific Editor
Journal of Cell Biology

Full Revision

Manuscript number: RC-2022-01392R

Corresponding author(s): Ilan Davis

1. General Statements

We thank the reviewers for their well informed and constructive comments. We have addressed every comment in detail, as explained below, and have revised the manuscript accordingly. The reviewing process has been excellent and has improved the manuscript in our view considerably.

In summary, here are some of the key improvements we feel we have made in response to the reviewers comments and suggestions:

- A) Additional data and improved description of Signal to Background
- B) Improved data and description of Dlg1::YFP and wild-type lines
- C) Explanation of some lines in some cells where protein is present but no mRNA detected
- D) Discussion of the possibility of protein secretion causing discordance
- E) Further information for each of the 200 genes in addition to molecular function
- F) Potential generality of the approach
- G) Issue of disruption of gene function by the YFP insertion
- H) Further analytical data on the issue of discordance between protein and mRNA expression
- I) Discussion of the issue of whether localised transcripts are translated
- J) Added supplementary figure for Indy::YFP
- K) Additional clarification of issue of protein trafficking
- L) Improvement of visibility of some colours in figures and “zoomed in” panels
- M) Corrections and improvements of all of the minor points made, especially by reviewer 3.

Furthermore, we made following corrections to the revised submission:

- A) Correction of a duplicated image in our Zegami database and its associated scorings
- B) Contrast improvements for a few images in our Zegami database
- C) Up to date analysis of protein and mRNA localisation in the NMJ glia (Supplementary Figure 3)

In addition to a clean revised manuscript, a marked up version of the manuscript has been submitted to indicate where we have made changes to the text and figures (in highlighted yellow text, also shown below in the relevant comments).

Full Revision

Reviewer #1 - - - - -

Major 1: For the wildtype CS flies, there is no YFP mRNA signal in neuroblast region and how about YFP mRNA signal in MB, OL VNC and NMJ regions? What is the criterion of setting laser power and gain for the mRNA level of 200 genes? Is it difficult to distinguish background and true signal of the mRNA in different area?

This is a good point about background intensity levels (from non-specific binding of the YFP smFISH probe) across different tissue regions. We thank the reviewer for raising it. The signal / background ratio decreases with depth in all of the tissues, with superficial cells displaying similarly high signal:background in the CNS and NMJ, while signal:background in neuropil regions of the CNS are slightly lower. To address this point, we have included a new supplementary figure to show background fluorescence of the smFISH probe across all regions of the CNS and NMJ (Figure S1A). We would also like to point out that the data we present is of extremely high quality considering we display data in more than 1300 figures each of which is in thick tissues, an imaging challenge in itself.

To address the point about image acquisition settings, we have included the following additional information in the Methods section (Page 17):

“Consistent image acquisition settings (laser power, detector gain, pixel dwell time or camera exposure) were used for experimental and control experiments. Acquisition settings were optimized to achieve fast imaging and high signal:background for each instrument.”

We have added a further explicit explanation to the manuscript referring to previous publications, regarding the discrimination of true mRNA signal over background:

Our YFP probe set produced punctate signals typical of individual transcripts that were diffraction-limited spots of uniform intensity and 3D fluorescence intensity distributions (Figure 1D-E) (J. S. Titlow et al. 2018; Raj et al. 2008; Yang et al. 2017). These consistent characteristics of the punctae allows us to easily distinguish true single molecules from discrete background fluorescence shapes that are either larger than diffraction-limited spots or have lower intensity than the single molecules (Figure 1E-N, S1A).

Major 2: Would the insertion of YFP affect gene expression? Compared to CS in Fig 1K, the *dlg1* mRNA signals in *dlg1::YFP* line (Fig 1F) increases a lot. I do not know if this phenotype happens only in this area. So could you show some other regions for *dlg1::YFP* flies.

Full Revision

This is a good point raised by both Reviewer #1 and Reviewer #2 (Major point 1). We agree that a proper quantification of the effect of YFP-insertion will bolster our conclusion, highlighting the utility of protein-trap collections for systematic analysis of post-transcriptional regulations. To address this point, we have provided quantifications of *dlg1* transcript and Dlg1 protein expression in the CNS and NMJ and compared the levels between Dlg1::YFP and wild-type lines. Our results suggest that the YFP insertion does not affect the transcript and protein expression levels of Dlg1 gene. The new figure visuals reflecting our quantification results are presented in Figure S1B-E.

Major 3: Is the *dlg::YFP* homozygous available? Among 200 gene trap lines, how many of them can be homozygous?

This point has been raised by both Reviewer #1 and Reviewer #3 (Major point 1). The *dlg1::YFP* (CPTI-002569) line used for the control experiments is homozygous. However, not all of the other YFP insertions are homozygous viable. Out of the 200 lines we screened, 131/200 (65.5%) insertions are homozygous viable, whereas 69/200 (34.5%) are homozygous lethal or are unknown. We have addressed this caveat in the Methods section (Page 16) with the following statement:

"While the majority of YFP insertion lines are homozygous viable (65.5%, 131/200), those that are not homozygous viable were kept over balancer chromosomes."

We also performed an additional analysis and added the following text and Figure:

To assess whether homozygous lethal YFP insertion leads to an overall reduced level of gene expression, we compared the number of nervous system compartments with YFP-fusion protein or mRNA expression between homozygous viable and lethal lines from our scoring. The analysis revealed homozygous viable and lethal lines show comparable numbers of nervous system compartments that express either protein or mRNA (see below, Wilcox test). Therefore, our data suggests the homozygous lethality is unlikely to skew our expression scoring survey.

Major 4: Have you tried to investigate the mRNA and protein localization in adult brains?

Yes, in a related study, we demonstrated that this approach also works in the adult brain (Mitchel et al., 2021, [DOI:10.7554/eLife.62770](https://doi.org/10.7554/eLife.62770)). A systematic analysis of protein and mRNA expression patterns in the adult brain would be highly interesting and is certainly possible, however it is beyond the scope of the manuscript. Nevertheless, we address this point by citing our previous work in the Discussion section as follows:

In a related study, we demonstrated that our approach can also be applied to the adult fly brain, where we characterize protein and mRNA expression of CamKII and five other well-known synaptic genes in a specific mushroom body output neuron (Mitchell et al. 2021).

Major 5: In Fig 3C, the authors claimed in MB or OL soma regions, some genes are protein expression only but no mRNA present. I wonder how do you explain this phenotype in soma.

Our favoured explanation is that protein is more stable than mRNA. Therefore, after the mRNA is translated, it could get degraded while the protein is still present in the cell. We have added the following text in the Results section (Overview of the Screen Results) to address this point:

We hypothesise that this expression pattern could be established by selective protein transport to the synapse, or higher protein stability that allows it to persist after the mRNA is degraded. he latter could also explain the presence of protein where there is no mRNA in the soma (Figure 3C).

Major 6: Since sgg mRNA localize to both sides of NMJ, would KCl stimulus affect sgg mRNA amount and localization in muscle?

That is an interesting question. The data in Fig. 8I-J show that there is no additional Sgg::YFP protein accumulation at the muscle post synaptic density in response to KCl stimulus. It's been shown elsewhere (Ataman et al., 2008, [DOI:10.1016/j.neuron.2008.01.026](https://doi.org/10.1016/j.neuron.2008.01.026)) that Sgg protein translocates to the muscle nucleus in response to KCl stimulus. Determining whether that mechanism requires translation of new protein would require a complete new study with translational analysis and would distract from the message of the current study.

Reviewer #2 - - - - -

Major 1: Although the group is using an established and published set of gene traps, it would be good to confirm protein expression for same gene to increase confidence or provide more details on how is known that the YFP insertions do not affect mRNA stabilization or

Full Revision

transcription or protein expression/localization. For example in Figure 1 F' versus K it is unclear why in the DlgYFP insertion there are more Dlg in situ signals than are observed in and around a neuroblast as compared to the wild type control. From the description provided these appear to the maximum intensity images. Is this due to background or an effect of the YFP insertion itself? Because of the increased level of expression is there a feedback loop of the protein regulating the mRNA expression? If had expression of Dlg protein in this figure would also confirm the YFP insertion mirrored the endogenous and it would be easier to discern if there were any changes in the number of Dlg mRNA molecules present. As this was the proof of principle example for the screen this information would increase confidence in the remainder of the data presented. AS an important part of the screen is looking at the potential for post transcriptional regulation this is an important factor to address.

Thank you for the valuable suggestion. We agree with the reviewer that the comparison of *dlg1* transcript levels would provide a valuable control. This point was raised by both Reviewer #1 and Reviewer #2. Please see [Reviewer #1 - Major point 2] for our response.

Major 2: Will this pipeline capture information on whether is secreted (contain a signal regulatory peptide) or not as then would expect to be discordant. This should be clarified or commented on.

This is a good point. We have added the following text to point out the possibility that secreted proteins, particularly in synaptic neuropil regions, could result in the target cell compartment containing protein in the absence of mRNA (not as a consequence of post-transcriptional regulation):

Pg. 13- Another point to consider is that secreted proteins could appear to be localised at a target cell where the mRNA is not expressed. This phenomenon was not observed for neuroblast lineages in the current study, but could not be ruled out for dense synaptic regions.

Major 3: General molecular function is listed in supplementary table 1 but will other types of information be able to be correlated from datasets or databases as well.

In addition to the general molecular function listed in supplementary table 1, the Zegami resource published with this manuscript is linked directly to Intermine and FlyBase.

Pg-6 "Moreover, the dataset lists other genes with known protein trap insertions and links directly to Intermine (Smith et al. 2012) and FlyBase (Larkin et al. 2021), which extends the utility of those resources."

In the text, we have noted that the Python pipeline is easily generalizable to other datasets:

Pg-6 "We designed and built a pipeline that is easily generalisable to other model organisms and data repositories."

Full Revision

We have also provided a link to a user manual that describes how to implement Zegami and the associated Python codes:

Pg-20 "All Python code is freely available at the GitHub repository: <https://github.com/ilandavislab/Annotate.OMERO.Fig> (copy archived at Zenodo here). A Zegami instance can be accessed here and the accompanying data are available in the Supplementary material. Guidance on how to use Zegami and browse the collection can be accessed here."

Minor 2: Could include more description on the generation of the supplementary tables and analysis of the tables. I could not find any description/legend which made analysis of some of the tables more difficult.

To address this point we have now included legends for each of the Supplementary Tables.

Minor 3: The data set was trained on a known set of data (analyzed by experts). It would be interesting to see what it could do with a novel set of genes in the context of post transcriptional regulation, but that is beyond the overall scope of this manuscript.

We agree- the approach is likely to be generalizable to novel gene sets and different physiological contexts, including post-transcriptional regulation (beyond the scope of the current manuscript). In the Results Section we have noted the following (which we feel is sufficient to address this comment):

"We designed and built a pipeline that is easily generalisable to other model organisms and data repositories."

Reviewer #3 - - - - -

Major 1: The approach relies on gene traps that often fail to be made homozygous, presumably due to deleterious function of the YFP insert. This is an obvious limitation of the study, which the authors address, but do so insufficiently by only analyzing a single case Dlg1. The authors should report how many of the 200 YFP-traps can produce viable homozygous animals, whether phenotypes can be observed, and any other relevant information to assess the functional properties of the tagged genes.

Thank you for requesting further information on homozygous viability of the YFP-trap collection. This point was raised by both Reviewer #1 and Reviewer #3. Please see **[Reviewer #1 - Major point 3]** for our response.

Full Revision

Major 2: The term "discordant" is used for non-congruous RNA/Protein levels in soma and distal processes, and sometimes the two are analyzed in the same figure (e.g Fig 3A). When it is stated that 98% of genes are discordant, this is an over-simplification as what the authors describe as "discordant" is expected to occur frequently in the distal process, but less often in the soma (which is what the authors find when presenting the data for individual compartments - Fig 3B-C). This is confusing because the observation means completely different things in the two compartments, though both are interesting to describe. These analyses, and their interpretation, should be kept separate.

This is a fair point raised by the reviewer. To address this point we have: (i) re-grouped the heatmap in Figure 3A to highlight the differences between intracellular and intercellular RNA/Protein discordance. (ii) We have also revised the text in the Results Section (Overview of the Screen Results) as follows:

The data in Figure 3A are grouped to distinguish intercellular discordance (between cells) from intracellular discordance (within a single cell), which are likely to arise from distinct mechanisms. Intercellular mRNA/protein discordance primarily occurs through differences in transcription, translation, or degradation rates between different cells. In contrast, intracellular mRNA/protein discordance is likely to be dependent on transport and localised translation, as well as translational repression. The two types of discordance were observed at similar frequencies in our dataset (51% intercellular discordance, 49% intracellular discordance).

Major 3: There is not enough emphasis placed on the cell-type specific regulation of RNAs. There are very few studies that have investigated how localization of individual RNAs changes in different cell types or regions of the nervous system, and the authors find that this is quite prevalent. Therefore, the rather superficial analysis of these data fails to take advantage of a major strength of the data. For example, for the discordant genes that differ in neuropil localization between different regions of the CNS, what types of molecules do they encode, what is their function in neurons (if known), and why might they be required locally in one region of the CNS but not the other?

We appreciate that the Reviewer recognizes the power of comparing RNA localization patterns across different brain regions (Figure 5R). We reported on a common set of synaptic mRNAs that encode nuclear proteins across the different regions of the nervous system. In an effort to address the Reviewer's suggestion, we have also now performed GO-enrichment analysis on region-specific patterns of expression. Those analyses are included in a new supplementary figure (Figure S4).

The following text has been added to the Results Section to explain the outcome of the analysis:

To gain further insight into which molecular functions are cell-specific or common across all three synaptic compartments we performed GO enrichment analysis of genes

with discordant RNA and protein expression (Figure S4). Discordant expression across all synapses were terms that are obviously related to mRNA localization and asymmetric function, such as cell-cell junction assembly, apical cytoplasm and cell periphery. Surprisingly, the genes with discordant expression in specific compartments were mostly enriched for unique functional terms related to development. This suggests that local expression of a common set of genes supports synaptic function while a cell-specific repertoire of local transcripts guides synapse development.

Major 4: The authors conclude that mRNA and protein co-localization in glia processes shows that mRNA localization makes a major contribution of the proteome in processes. However, there is not enough evidence for such conclusion since neither translation of these mRNAs nor lack of protein trafficking from the somas was shown.

This is a good point, as we do not show any functional evidence from our own data. Nevertheless, the significant role of mRNA localisation in shaping the local proteome and performing proteostatic regulation has been studied in detail by others (Zappulo et al., 2017, von Kugelgen and Chekulaeva 2020 Giandomenico et al., 2022). To address this comment, we have: (i) clarified the text by including the citation of these publications, and (ii) qualified our claim in the manuscript that mRNA localization is a major contribution of the proteome in neurite or glial processes. The following has been added:

Together, analysis of mRNA and protein expression in glial processes of the CNS and PNS shows potential contribution of mRNA localisation to the proteome in that compartment (Supplementary Table S3) (Giandomenico, Alvarez-Castelao, and Schuman 2022; von Kugelgen and Chekulaeva 2020; Zappulo et al. 2017).

Zappulo et al., 2017, [DOI: 10.1038/s41467-017-00690-6](https://doi.org/10.1038/s41467-017-00690-6)
von Kugelgen and Chekulaeva 2020 [DOI: 10.1002/wrna.1590](https://doi.org/10.1002/wrna.1590)
Giandomenico et al., 2022, [DOI: 10.1016/j.tins.2021.08.002](https://doi.org/10.1016/j.tins.2021.08.002)

Major 5: An important caveat of this technique that should be discussed is the lack of knowledge about the translation of these mRNAs, if the mRNA that is being detected is the same as the one that is translated. While the authors emphasize the discordance between mRNA and protein localization, it is not possible to know whether these mRNAs are being translated where they are found, e.g. soma vs neuropil. Moreover, there are many examples (e.g. BDNF) where the isoform influences the subcellular localization of the mRNA. There is no way of studying the isoforms here, and we could be looking for a different mRNA isoform localized to a specific compartment compared to the protein. These points must be discussed.

Full Revision

We agree with the reviewer that our method does not provide information on whether the detected mRNA is being specifically translated in time and space. To address the point, we have revised the manuscript to address this more explicitly (see below). Elucidating the relative contribution of localised mRNA in shaping the local proteome is not a trivial task and it is being actively investigated in the field. However, we believe our dataset provides a unique high-resolution map of transcripts that are potentially regulated at post-transcriptional and translational levels. It would be promising to follow up the 'discordant' genes identified from our survey using experimental methods that are able to track mRNA-ribosome associations (e.g. TRICK) in future studies. To address this point, we have added the following text in the Discussion Section as requested by the reviewer.

“... because it is not clear from our data whether these genes are more translationally active and/or translationally repressed across different stages of cell differentiation. Identifying the association with ribosomes and the trans-acting factors of specific mRNAs at different stages of development will be necessary to fully understand how these genes are regulated to influence neural differentiation and synaptic transmission (Richer, Speese, and Logan 2021; Halstead et al. 2016). In addition, alternative splicing and alternative polyadenylation may influence the fate of mRNA transport and translation in specific cases, which would not be detected with our method (Tian and Manley 2017).”

We thank the reviewer for highlighting their point about mRNA isoforms. Our preliminary analysis indicates that 71% of the screened genes have constitutive YFP-insertions (i.e. YFP-cassette traps all mRNA isoforms). However, we agree that our approach cannot discriminate between specific cases where protein produced from an mRNA isoform is trafficked and co-localises with another mRNA isoform that did not give rise to that protein. The added text above also addresses this point about isoform specificity.

Minor suggestions:

• The authors should identify GO terms to understand what types of molecules are subjected to RNA regulation. They provide a supplementary table for all genes, but it would be useful to have a chart showing the proportion of different GO terms represented in the overall gene set, genes that show cell-specific regulation, genes that show neuron vs glia specific regulation, etc.

The analysis that the reviewer suggested in the following table below. Proportion of GOSlim terms is a metric that we used to show functional coverage of the screen collection. It is not immediately clear what one learns by applying this metric to cell-specific gene sets. The percentage of GOSlim terms in a set of genes is correlated with the number of genes in the set.

Full Revision

Gene set	# of genes	GOSlim Coverage (% of terms)
whole collection	200	90
neuro-specific regulation	179	87
glia-specific regulation	56	61
Neuroblast cell-specific regulation	43	66

We have revised the text to highlight this table more clearly.

- **"However, post-transcriptional regulation can also manifest itself within a cell, so that a protein is localised to a distinct site from the mRNA that encodes it". While subcellular RNA localization may represent a regulatory layer, I do not agree that proteins that function in the cell at a different location than their translation site represents regulation per se. Many such cases exist for proteins that are trafficked!**

We agree. We have revised that sentence to remove the mention of post-transcriptional regulation to avoid confusion. This minor change now hopefully makes the existing sentence following the sentence in question clearer by mentioning protein transport and a review that provides examples:

"Many mechanisms can lead to intracellular protein and mRNA discordance, including localised translation, mRNA degradation or intracellular transport of protein or mRNA (Mofatteh and Bullock 2017)."

- **"The majority of individual puncta appearing in the dlg1::YFP line (51% in the brain, 64% in larval muscles". Why is the agreement between YFP and endogenous FISH so low? Do many individual RNAs fail to hybridize? This should be discussed.**

We agree that the co-detection numbers from that single experiment were a little low. Although sufficient for the purpose we used the control, it represented an over cautious conservative estimate. We have since repeated the experiment and achieved a higher percentage of co-detection in both tissues, which we feel are more representative of the data presented. These values have been updated in the manuscript.

(85% in the brain, 78% in larval muscles (Figure 1E"))

- **"However, one gene, indy, is highly transcribed in neuroblasts and a single ganglion mother cell before it is rapidly shut off (Figure S1A)". This figure does not exist. Where are the data?**

This data was present in our Zegami instance (amongst >1300 figures). To address this point and make it easier to find, we have now added image panels related to Indy expression in the Figure S3.

- **"We only observe a minority of localised axonal mRNAs that lack the protein they encode at the axon extremities, in contrast to our findings in the mushroom body, optic lobe, and ventral nerve cord neuropils"**

Full Revision

These results are not contrasted, as in all neuropils the minority of localized mRNAs are those lacking their corresponding proteins. For example, 9% in NMJ vs 7.5% in OL neuropil according to Fig. 1B. What is conflicting with the conclusion?

This is a good point. We have revised that statement to say that the NMJ axon terminal has fewer mRNAs than any of the other peripheral neural compartments. To address this point we have revised the manuscript:

~~We only observe a minority of localised axonal mRNAs that lack the protein they encode at the axon extremities,~~ **The NMJ axon terminal compartment contained far fewer mRNA species than mushroom body, optic lobe, and ventral nerve cord neuropils.**

- **"These results suggest that motor axons are more selective than the other neuronal extensions in the mRNAs that are transported over their very long distances from the soma to the neuromuscular synapse"**

The current literature says that the same mechanism (cis-elements) is used to transport mRNAs to subcellular compartments, which would be inconsistent with the idea of motor axons being "more selective" than other neurons for the same mRNA, but just a result of fewer mRNAs being found in motor neurons: 34.% of the mRNAs are found in motor neurons soma vs 83% in OL soma, 86.5% in VNC soma, and 70.5% in MB soma. To get to this conclusion, the authors should show that mRNAs previously found in the neuronal extensions of other neurons are not found in the axons of motor neurons but are still expressed in their somas. They might want to suggest different RBPs involved in the transport or discussing the very long distance they need to travel which can influence their detection in the tips.

We agree with the Reviewer that RBP repertoire and distance from the soma are plausible explanations for why fewer mRNA species are detected at the NMJ axon terminals. We've updated the text to include these explanations.

~~We only observe a minority of localised axonal mRNAs that lack the protein they encode at the axon extremities,~~ **The NMJ axon terminal compartment contained far fewer mRNA species than mushroom body, optic lobe, and ventral nerve cord neuropils. These results suggest at least one of three plausible mechanistic explanations, none of which are mutually exclusive. One is that the motor axons have a different repertoire of RNA binding proteins restricting entry into the axon (Martínez et al. 2019) and/or transport to the axon terminals. Another possible explanation is that only the most stable mRNAs are able to avoid degradation across the extremely long distance from the soma to the NMJ. Finally, transcripts that are detected in the motor axon terminals could have distinct localization signals.** ~~These results suggest that motor axons are more selective than the other neuronal extensions in the mRNAs that are transported over their very long distances from the soma to the neuromuscular synapse.~~

Full Revision

We have also followed the Reviewer's advice to search our dataset for mRNAs present in other neuropils and motor neuron soma that are not present in motor axon terminals, as these mRNAs would be candidates for cell-specific regulation. In mushroom body neuropil alone there are 54 mRNAs that are also present in motor neuron soma but not axon terminals.

- **Figure 1. Experimental approach summary**

- o **Some colors do not show well and should be changed, e.g: grey in Fig. 1A, and Fig. 1B probe sites indicated in light blue and pink within the introns of *dlg1*.**

Thank you, colours have been updated to improve visibility.

- o **Fig. 1E': There appears to be a large discrepancy in co-detection % for CNS and muscle in the graph judging by the size of circles, yet in the text, it is stated that there is average of 51% and 64% in the two, respectively. I don't see any green circles with over 25% agreement in the graph. Are the colors correct here?**

We have re-assessed the co-detection percentage for both CNS and NMJ compartments. The average co-detection levels between YFP and *dlg1* smFISH probes are 85% and 78%, respectively for CNS and NMJ compartments. The Figure 1E' has been updated to reflect our quantification results.

- o **Fig. 1D-I: It's difficult to identify where the zoomed panels come from. E has its own square (indicating zoom in E'). Please make this square dashed or a different color in E so it is clear F and G do not come from there.**

To address this point, the zoomed panels have now been given different colors to make them more visible.

- o **Comparing Fig. 1F vs K: Why does there appear to be so much more *dlg1* mRNA in the YFP-tag condition? If this is due to selection of imaging area, please choose a more similar region to image so the RNA levels are comparable. Otherwise it indicates the YFP-tag line has more RNA expression, which is likely not the case.**

The Figure 1D-N has been updated to show a consistent number of Z slices for Dlg::YFP and Wild-type genotypes. Our quantification of *dlg1* spots between Dlg::YFP and Wild-type nervous system indicates the *dlg1* transcript levels are comparable (Figure S1D-E), and that the YFP insertion does not affect expression levels

- **Figure 3. Overall summary of results**

o Figure 3A: Soma/Neuropil/muscle should be separate or at least ordered such that they are next to each other to facilitate direct comparison of genes in the same region of the cell in neurons from different CNS areas. Why are glia not included in this summary? A third color should be used to indicate when there is neither mRNA nor protein expression.

We updated Figure 3A to indicate when there is neither mRNA nor protein expression using another colour.

Localisation of RNA and protein in Glia were assessed using an independent method from our scoring method presented in Figure 3, as it was much harder to carry out without a specific glial marker. The glial cellular compartment was initially approximated based on their typical morphology in the NMJ terminals and stereotypical protein expression in the central nervous system, and the glial localised RNA expression was cross-checked in some specific examples using fluorescently labelled glial lines. Therefore, we decided not to include it in the same summary table as the neuronal compartments, however, the full glial localised gene expression scoring is provided in the Supplementary File 3.

• Figure 4. Neuroblast lineage analysis

o Is clustering around the NB sufficient to determine lineage relationship? There seems to be other neurons around the NB.

To address this point, we have added the following text to explain how the anatomy surrounding a NB enables accurate definition of lineage relationships:

“NB lineages are typically grouped into a single compartment and surrounded by glial cells providing a glial niche, which can allow the NB lineage to be identified unambiguously. We performed our scoring in these clearly identifiable NB lineages.”

o More examples should be shown for the post-transcriptional category, as it is the most interesting category, and there are many different possible outcomes. Are there cases of transcriptional control and post-transcriptional regulation? Are there cases where the youngest neurons (closer to the NB) in the progeny are expressing the protein while the oldest are not? If not, could this be an artifact from a slow translation and the protein being detected only after building up in the cell? Top1 protein (Fig. 4D) seems to be less expressed in the youngest neurons.

Thank you for the comment. Indeed, there are too many permutations of mRNA:protein ratio that could be present across the neuroblast lineages. We have therefore made the complete dataset available for anyone to browse and analyze if interested (in a Zegami instance containing >1300 figures).

Full Revision

We agree with the Reviewer's observation that Top1 appears to be expressed at a lower level in the youngest neurons, and that slow translation is a plausible explanation. However, we have respectfully chosen not to add this speculation to the text.

o "The transcription rate of these genes, as indicated by the relative intensity of smFISH nuclear transcription foci, is similar across the neuroblast lineage, however protein signal is only detectable in a minority of the progeny cells (Figure 4E)".

Many nuclei lack clear large spots, but have small spots indicative of RNA; how is this interpreted? Do they lack transcription, or is this due failure of the smFISH to capture all transcription sites?

There are multiple interpretations of nuclei that lack transcription foci in cells that express cytosolic mRNA. The simplest biological interpretation is that transcription at that loci has been turned 'off' while there are still mRNAs present in the cytosol. Transcription is a 'bursty' process, and pre-mRNAs are quickly exported to the nucleus after transcription termination and splicing, therefore it is not surprising to detect cytosolic mRNAs in the absence of highly active transcription foci.

Detection of transcription foci is actually far more robust than cytosolic mRNA detection— there are multiple pre-mRNA in a diffraction-limited area as opposed to single transcripts in the cytoplasm— so it is unlikely that transcription sites would be undetectable by smFISH if single mRNAs in the cytosol are detected.

These points have been addressed in the MS with the following statement:

Some cells in Figure 4B and 4E have cytoplasmic mRNA in the absence of obvious transcription foci. The simplest biological interpretation is that transcription at those loci has been turned 'off' while there are still mRNAs present in the cytosol. and pre-mRNAs are quickly exported to the nucleus after transcription termination and splicing, therefore it is not surprising to detect cytosolic mRNAs in the absence of highly active transcription foci. It is also possible that the transcription foci were not captured in the optical section. Detection of transcription foci is actually far more robust than cytosolic mRNA detection— there are multiple pre-mRNA in a diffraction-limited area as opposed to single transcripts in the cytoplasm— so it is unlikely that transcription sites would be undetectable by smFISH if single mRNAs in the cytosol are detected.

Were transcripts actually counted to assess cell-specific differences? This should be possible with smFISH

Cell-specific differences were determined using a binary assessment of whether RNA or protein was present or absent. Counting transcripts is indeed possible, and would likely reveal subtle variations in the ratio of mRNA:protein in cells as they progress through different stages of differentiation, but it is too hard to achieve for so many genes, so is beyond the scope of this study..

Full Revision

- **Figure 5. RNA synaptic localization**

- o **A have global analysis comparison of all neuropil areas would be welcome in this figure.**

A global comparison of all neuropil areas is provided in Figures 3A-C.

- o **For Fig. 5J and 5N RNA appears scattered regularly throughout the entire panel area. How sure are the authors that this is not due to poor signal/noise? For example, perhaps too much probe being used for these targets.**

This is a good question. Based on our negative controls (Fig 1L) and examples that show similar levels of somatic RNA signal in the absence of neuropil RNA signal (Fig 5F), and the fact that we use the same smFISH probe against the same YFP insert for all of these lines, and the fact that the spots in the image show the hallmark distribution of uniform intensity and size throughout the cytoplasm with larger nuclear spots (transcription foci) we are quite confident that the RNAs present in the synaptic neuropil are not due to poor signal/noise.

- **Figure 6. RNA localization in glia**

- o **For Fig. 6B-G it is hard to tell if there is any overlap of the RNA and Glia. Maybe show multiple zoomed-in merged images and/or highlight the structures with lines that are present in all panels.**

To address this comment, we have overlaid the glial marker on the RNA data in Fig 6D and G.

- o **For Fig. 6L-O: How reproducible is this small amount of RNA puncta in the NMJ glia? Is this possibly biologically important?**

We found that 19 different genes express mRNA at glial processes of the NMJ, which indicates reproducibility across genotypes. We have multiple animals/NMJs for several of those genes. Moreover, we have confirmed peripheral glial mRNA expression for 18 of those genes in a follow up study (that will be published elsewhere later).

Assessing biological importance of localized mRNA is technically very difficult, and requires discovery and mutation of localization elements in the untranslated or coding sequence that disrupt transport without disrupting mRNA stability or translation. We are in the process of generating such mutants, but this level of characterization far exceeds the scope of the current study. What we can say is that the literature contains several examples of functional RNA localization in neurons that has a similar paucity of transcripts. Moreover mRNAs have been shown to undergo hundreds of rounds of translation, suggesting that a few molecules can have a substantial functional impact in a small subcellular compartment.

- o **Why do cartoons labelling subnuclear/perinuclear glia in Fig.6 and Fig.S6 show different localization?**

Full Revision

We updated figure S5 and the cartoons match now. Thank you to the reviewer for pointing this out.

o The cartoons seem to extrapolate from the data: While in Fig 6B-D, we see neither the big bright spot of transcription in the glial nucleus nor as many transcripts in the neuropil, they are both present in the cartoon. In Fig. 6E-G there is no indication of cortical glia soma nor the transcription spot only in glia nuclei.

We updated figure 6 so that the cartoons next to panels 6B-D and 6E-G match the panels respectively.

o "To assess glial localisation for the 200 genes of interest, we used a pan-glial gal4 driving a membrane mCherry marker (repo-GAL4>UAS-mcd8-mCherry) to learn the expression pattern of all glial cells, and then classified the pattern in the YFP lines (without the marker) based on knowledge of that expression pattern. We validated this approach by combining the RFP marker"

Did the authors use mCherry or RFP for these experiments? Also, the previous sentence is redundant.

Thank you to the Reviewer for spotting these issues. We used mCherry, and have updated the text and figures. The redundant sentence has also been updated to the following:

'To validate that the repo-GAL4>UAS-mcd8-mCherry indeed faithfully represented the location of glial cells throughout the nervous system, we assessed the overlap of the repo-GAL4>UAS-mcd8-mCherry labelling with the Nrv2::YFP insertion, one of the 200 lines which was already known to be a wrapping glial marker (Yadav et al. 2019)'

• Figure 7. RNA localization at neuromuscular synapse

o RNA for these genes seems far too spread throughout the muscle to draw any conclusions

It is true that mRNAs shown are expressed in both the axon and the muscle, however **the spatial resolution of the microscope combined with the clear delineation of axon and PSD compartments by the two markers provide sufficient discrimination to support the qualitative conclusion that molecules are present in either or both compartments.** We have added this statement (in yellow) explaining this point in the methods section.

o Also with so many RNAs distributed in the muscle, specific localization of RNA molecule to the precise PSD would have no conceivable benefit

This is a reasonable hypothesis. However, we can conceive of at least two benefits for having mRNAs localised specifically to the PSD:

Full Revision

- 1) Rapid and local production of protein in response to synaptic activity-it is not currently known how long it takes for proteins to translocate across the subsynaptic reticulum, therefore it could be advantageous to have protein synthesised directly within the SSR
- 2) Production of PSD-specific post-translational modification (PTM) signatures- Though it has been shown that PTMs are required for some proteins to be inserted into the PSD and form complexes with other scaffolding proteins, it is not currently known where the PTMs occur. Translation specifically within the PSD could enable compartment-specific regulation of PTMs.

The current data do not rule out the possibility that synapses require translation specifically within the PSD. We have revised the discussion to add a short statement about this interesting point.

Given the high abundance of mRNA in the muscle cytoplasm and in close proximity to the synapse, what would be the benefit of localized mRNA? There are at least two probable functions, one is rapid and local production of protein in response to synaptic activity. It is not known how long it takes for proteins to translocate across the subsynaptic reticulum (SSR), therefore protein synthesis directly within the SSR may be required to produce highly localized signaling molecules on the appropriate timescale. Another potential function is the production of PSD-specific post-translational modification (PTM) signatures. Though it has been shown that PTMs are required for some proteins to be inserted into the PSD and form complexes with other scaffolding proteins, it is not currently known where the PTMs occur. Translation specifically within the PSD could enable compartment-specific regulation of PTMs.

o I suggest drawing lines around the protein expression to facilitate visualization of the mRNA localization for panels B, F and J. It is especially hard to conclude anything from panels B and F.

We have now added markers to Figure 7B,F, and J to facilitate visualization, as the reviewer suggested.

o Light grey with white dots is hard to see in the cartoons

We have now made the background in the cartoons darker to increase contrast.

• Figure 8. Role of khc and activity in sgg localization

o Presumably there is a huge number of developmental problems associated with this mutant that could cause decrease in sgg localization

o If the authors include this, then they should characterize the mutant NMJs: what is the change in size, synapse number, etc..

Full Revision

This is a good point. However, the *khc* mutants are already known to show synapse morphology phenotypes (Kang et al., 2014) and the *khc*²³/*khc*²⁷ transheterozygous allele has previously been used to assess localization defects at the larval NMJ (Gardiol and St. Johnston, 2014). Moreover, our manuscript (Figure 8) focuses on post-developmental stimulus-dependent processes, rather than cellular-level synapse developmental parameters with this mutant. The reviewer correctly points out that the *khc* developmental phenotypes are likely to have other secondary defects as a result of impaired microtubule transport. The purpose of that mutant was to assess the molecular-level question of whether microtubule-based transport is required for *sgg* mRNA localization at the axon terminal. The consequences and exact mechanism of disrupted transport are beyond the scope of this study. To address this point explicitly, we have:

1. Revised the manuscript to quote more explicitly and clearly the developmental *khc* phenotype.
2. Revised the manuscript to explain the difference between the developmental role of *khc* and role in the transport of *sgg* specifically to the axon terminal.
3. Revised the manuscript to explain more explicitly the limitations of this mutant.

The following text was added:

*While it is important to consider that the mRNA localization phenotype occurs in the context of abnormal synaptic and organism development in the trans-heterozygous *khc* mutant background (Gardiol and St Johnston 2014; Kang et al. 2014), the result strongly indicates that kinesin-1 based transport is required for *sgg* localisation at the NMJ.*

o Is there more *sgg* accumulated in soma as a result of less transport? Is *sgg* being expressed at the same level?

This is an interesting question that has come up before. Fortunately, we have *sgg* smFISH data from motor neurons in the VNC of wild type and *khc*²³/*khc*²⁷ mutants. Quantification of *sgg* mRNAs in motor neurons did not show an accumulation of *sgg* mRNAs as a result of less transport. *sgg* mRNA levels were actually slightly lower in the *khc* mutants. We have included those data in Figure 8S, showing representative image data from both genotypes and the quantification. We have also reported the result in the Results section of the text as follows:

*We asked if the absence of *khc* transport causes *sgg* transcripts to accumulate in the motor neuron soma, however there was no evidence of increased *sgg* in *khc*²³/*khc*²⁷ mutant VNCs (Figure S6).*

o Fig. 8F-H: Why is Dlg1 accumulated in the entire axon, not just the presume synapse?

Dlg1 is present throughout the subsynaptic reticulum, which envelops the synapse as well as extrasynaptic regions between boutons. Our results are consistent with previously published

Full Revision

anti-Dlg1 staining between boutons, see Menon et al., 2013- WIREs Dev Bio for an authoritative review.

o Fig. 8J: Why is sgg signal occurring in circles disconnected from the main axon? The authors should show a different image

Good point. It is difficult to see the neurite connecting the bouton to the axon branch. We have replaced the image with an example that shows highly expressing boutons with more visible connections.

3. Description of the revisions that have already been incorporated in the transferred manuscript

Please insert a point-by-point reply describing the revisions that were already carried out and included in the transferred manuscript. If no revisions have been carried out yet, please leave this section empty.

Reviewer 1 - Minor 1: You claimed that Fig 1E shows high magnification image of the inset in D, but the scale bars are the same.

The scale bar in panel E has been corrected to match the magnification of the panel compared to panel D

Reviewer 1 - Minor 2: Figure 1 legend: K-N, are the images individual channels shown in E? Or in J?

Those images are indeed the individual channels in J, not E. This typo has been corrected.

Reviewer 1 - Minor 3: In Fig 2A, optic lobe neuropil and VNC neuropil are mislabeled.

We apologise, the images were mislabelled. The Figure has now been corrected.

Reviewer 1 - Minor 4: Only one panel has scale bar in Fig 4.

Scale bars have been added to the other panels in Fig. 4 and scale for top1 was corrected.

Reviewer 1 - Minor 5: What is Fig 5B'and F'? You should describe them in the Figure legends.

Panels 5B' and 5F' are high magnification images of panels 5B and 5F. The following text has been added to the figure legend for clarification:

Full Revision

"(B',F') High magnification images of the neuropil regions outlined by a white box in panels B and F."

Reviewer 2 - Minor 1: On page 9 refer to Figure 6S which I think is supposed to be Figure S6. In text refer to an example of gli but show gs2 in the figure so it is unclear what is being referred to or shown.

The text should refer to *gs2*, which is shown in the updated Fig. S5. Both typos have been corrected.

Reviewer 3 - Minor comments

• The authors should be consistent about calling perineurial or perineural glia (both correct) in their images and text.

Images and text are now consistent, glia referred to as either perineural or subperineural.

• Figure 2. Analysis pipeline overview

o The lines for the first two zoomed panels are switched: The optic lobe is going to VNC and vice-versa.

Corrected.

o "Compiling all the information together shows that there are that 196/200 or 98% of the genes show discordance between RNA and protein expression"

However, 5 genes shown in Fig. 3A do not show "discordance": CG9650, cup, Lasb, rg, and vsg!!

The numbers and percentages have now been updated to reflect the correct values. There are 195 genes that show discordance or 97.5% of genes.

o "Surprisingly, another 59 transcripts are present at synapses without detectable levels of protein (Figure 5E-H)"

This text does not correspond to Fig 5E-H but 5I-L. Where is the text about 5E-H?

The text indeed corresponds to 5I-L. We corrected the typo. We also updated the corresponding text to describe Figure 5E-H.

We revised the Figure 5 legend so that the numbers correctly represent our quantifications.

o Fig. 5R is not cited in the text.

Figure 5R is now cited in the text.

November 15, 2022

RE: JCB Manuscript #202205129R

Prof. Ilan Davis
University of Oxford
Department of Biochemistry
South Parks Road
Oxford OX1 3QU
United Kingdom

Dear Prof. Davis,

Thank you for submitting your revised manuscript entitled "Systematic analysis of YFP gene traps reveals common mRNA/protein discordance in neural tissues." We would be happy to publish your paper in JCB pending final revisions necessary to meet our formatting guidelines (see details below).

A. MANUSCRIPT ORGANIZATION AND FORMATTING:

- 1) Text limits: Character count for Tools is < 40,000, not including spaces. Count includes title page, abstract, introduction, results, discussion, and acknowledgments. Count does not include materials and methods, figure legends, references, tables, or supplemental legends.
- 2) Figure formatting: Tools may have up to 10 main text figures. Scale bars must be present on all microscopy images, including inset magnifications. Please add scale bar for Fig6 panels M/N/O as well as the inset magnification in panel O.
- 3) Statistical analysis: Error bars on graphic representations of numerical data must be clearly described in the figure legend. The number of independent data points (n) represented in a graph must be indicated in the legend. Statistical methods should be explained in full in the materials and methods. For figures presenting pooled data the statistical measure should be defined in the figure legends. Please also be sure to indicate the statistical tests used in each of your experiments (both in the figure legend itself and in a separate methods section) as well as the parameters of the test (for example, if you ran a t-test, please indicate if it was one- or two-sided, etc.). Also, if you used parametric tests, please indicate if the data distribution was tested for normality (and if so, how). If not, you must state something to the effect that "Data distribution was assumed to be normal but this was not formally tested."
- 4) Materials and methods: Should be comprehensive and not simply reference a previous publication for details on how an experiment was performed. Please provide full descriptions (at least in brief) in the text for readers who may not have access to referenced manuscripts. The text should not refer to methods "...as previously described."
- 5) For all cell lines, vectors, constructs/cDNAs, etc. - all genetic material: please include database / vendor ID (e.g., Addgene, ATCC, etc.) or if unavailable, please briefly describe their basic genetic features, even if described in other published work or gifted to you by other investigators (and provide references where appropriate). Please be sure to provide the sequences for all of your oligos: primers, si/shRNA, RNAi, gRNAs, etc. in the materials and methods. You must also indicate in the methods the source, species, and catalog numbers/vendor identifiers (where appropriate) for all of your antibodies, including secondary. If antibodies are not commercial, please add a reference citation if possible.
- 6) Microscope image acquisition: The following information must be provided about the acquisition and processing of images:
 - a. Make and model of microscope
 - b. Type, magnification, and numerical aperture of the objective lenses
 - c. Temperature
 - d. Imaging medium
 - e. Fluorochromes
 - f. Camera make and model
 - g. Acquisition software
 - h. Any software used for image processing subsequent to data acquisition. Please include details and types of operations involved (e.g., type of deconvolution, 3D reconstitutions, surface or volume rendering, gamma adjustments, etc.).

7) References: There is no limit to the number of references cited in a manuscript. References should be cited parenthetically in the text by author and year of publication. Abbreviate the names of journals according to PubMed.

8) Supplemental materials: Tools generally have up to 5 supplemental figures and 10 videos. You currently exceed this limit but, in this case, we will be able to give you the extra space. Please also note that tables, like figures, should be provided as individual, editable files. A summary of all supplemental material should appear at the end of the Materials and methods section. Please include one brief sentence per item.

9) eTOC summary: A ~40-50 word summary that describes the context and significance of the findings for a general readership should be included on the title page. The statement should be written in the present tense and refer to the work in the third person. It should begin with "First author name(s) et al..." to match our preferred style.

10) Conflict of interest statement: JCB requires inclusion of a statement in the acknowledgements regarding competing financial interests. If no competing financial interests exist, please include the following statement: "The authors declare no competing financial interests." If competing interests are declared, please follow your statement of these competing interests with the following statement: "The authors declare no further competing financial interests."

11) A separate author contribution section is required following the Acknowledgments in all research manuscripts. All authors should be mentioned and designated by their first and middle initials and full surnames. We encourage use of the CRediT nomenclature (<https://casrai.org/credit/>).

12) ORCID IDs: ORCID IDs are unique identifiers allowing researchers to create a record of their various scholarly contributions in a single place. At resubmission of your final files, please consider providing an ORCID ID for as many contributing authors as possible.

B. FINAL FILES:

Thank you for this interesting contribution, we look forward to publishing your paper in Journal of Cell Biology.

Sincerely,

Juergen Knoblich, PhD
Monitoring Editor
Journal of Cell Biology

Dan Simon, PhD
Scientific Editor
Journal of Cell Biology